# Discovering the Gems in Early Layers: Accelerating Long-Context LLMs with 1000x Input Token Reduction

## Abstract

Large Language Models (LLMs) have demonstrated remarkable capabilities in handling long context inputs, but this comes at the cost of increased computational resources and latency. Our research introduces a novel approach for the long context bottleneck to accelerate LLM inference and reduce GPU memory consumption. Our research demonstrates that LLMs can identify relevant tokens in the early layers before generating answers to a query. Leveraging this insight, we propose an algorithm that uses early layers of an LLM as filters to select and compress input tokens, significantly reducing the context length for subsequent processing. Our method, GemFilter, demonstrates substantial improvements in both speed and memory efficiency compared to existing techniques, such as standard attention and SnapKV/H2O. Notably, it achieves a $2.4\times$ speedup and 30% reduction in GPU memory usage compared to SOTA methods. Evaluation on the Needle in a Haystack task shows that GemFilter significantly outperforms standard attention, SnapKV and demonstrates comparable performance on the Long-Bench challenge. GemFilter is simple, training-free, and broadly applicable across different LLMs. Crucially, it provides interpretability by allowing humans to inspect the selected input sequence. These findings not only offer practical benefits for LLM deployment, but also enhance our understanding of LLM internal mechanisms, paving the way for further optimizations in LLM design and inference.

## 1 Introduction

Large Language Models (LLMs) have demonstrated impressive abilities (Wei et al., 2022; Bubeck et al., 2023) and found widespread application in various AI systems, such as ChatGPT (Schulman et al., 2022), Gemini (Anil et al., 2023), and Claude (Anthropic, 2024), and so on. They are also a fundamental component in building language-based AI agents that can orchestrate plans and execute complex tasks through interaction with external tools. A key requirement for many of these applications is the ability to process long-context inputs. This ability can also potentially eliminate the need of a retriever in retrieval augmented generation (RAG) (Xu et al., 2024a) or enhance its performance (Jiang et al., 2024c). Therefore, significant efforts have been made recently to build LLMs that support long context inputs. For instance, LLaMA 3.1 (Dubey et al., 2024), Mistral (Jiang et al., 2023a), and Phi 3.5 (Abdin et al., 2024) now support input sequences of up to 128K tokens, while Gemini can handle inputs of up to 1M tokens. However, processing such lengthy inputs comes at a substantial cost in terms of computational resources and time. Therefore, accelerating the LLM generation speed while simultaneously reducing GPU memory consumption for long-context inputs is essential to minimize response latency and increase throughput for LLM API calls.

One prominent optimization for fast text generation in decoder-only LLMs (i.e., using a causal attention mask) is the *KV cache*. Specifically, there are two phases involved in auto-regressive generation. Given a long context input, the first is the *prompt computation* phase, when the LLM computes the KV cache for all layers, storing the intermediate attention keys and values of the input tokens. Next, in the *iterative generation* phase, the LLM generates tokens iteratively using the pre-computed KV cache, avoiding redundant computations. GPU memory usage and running time scale linearly with the KV cache size, meaning that the computational is high for long inputs.

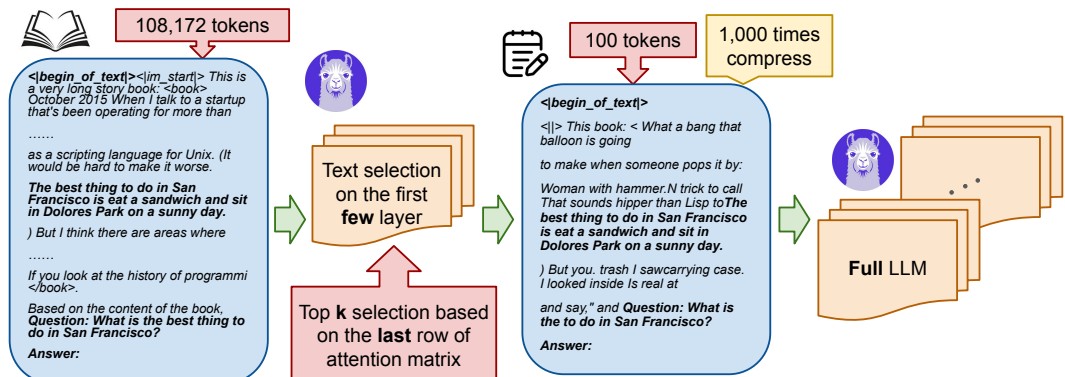

Figure 1: Illustration of our method GemFilter: generation with context selection based on early filter layers. We demonstrate a real Needle in a Haystack task (Section 4.1). The original input consists of 108,172 tokens, including the initial instruction, key message, and the query. In the first step, we use the 13th layer of the LLM (LLaMA 3.1 8B Instruct) as a filter to compress the input tokens by choosing the top $k$ indices from the last row of the attention matrix. Notably, the selected input retains the initial instruction, key message, and query. GemFilter achieves a $1000\times$ compression, reducing the input token length to 100. In the second step, we feed the selected tokens for full LLM inference using a standard generation function, which produces the correct output. GemFilter significantly reduces running time and GPU memory with negligible performance loss.

To reduce GPU memory usage and running time during the iterative generation phase, H2O (Zhang et al., 2023) and SnapKV (Li et al., 2024b) introduce static methods to compress/evict the KV cache. These techniques can shrink the KV cache size from 128K to 1024 with negligible performance loss, resulting in faster speeds and lower GPU memory consumption during the iterative generation phase. However, these methods do not improve the efficiency of the prompt computation phase, which becomes the dominant bottleneck as the input context lengthens. Thus, we ask:

*Can we accelerate the speed and reduce memory usage during the prompt computation phase?*

We observe that when serving a query, LLMs often find the necessary information in the early layers, even before generating the answer. Specifically, the relevant tokens can be identified using the attention matrix from these early layers (Figure 2), which we refer to as *filter layers*. Figure 1 provides a real example from the Needle in a Haystack task, where LLMs must find a small piece of information within a large context. For LLaMA 3.1 8B, we observe that the information needed to answer the query can be distilled from the attention matrix in any of the 13th-19th layers. Furthermore, LLMs explicitly summarize the required information in these filter layers. As a consequence, we only need to perform the prompt computation on a long context input for the filter layers, allowing us to compress the input tokens into a smaller subset (e.g., reducing from 128K tokens to 100), saving both time and GPU memory. We then feed the selected tokens for full model inference and proceed with a standard generation function. Algorithm 1 in Section 3 presents our method GemFilter.

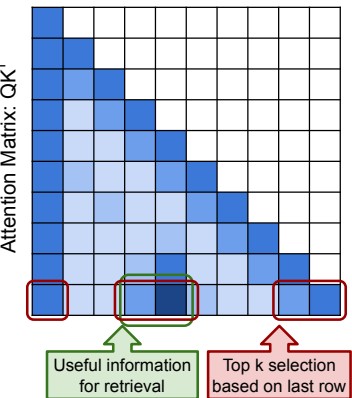

Figure 2: The last row of attention matrices in early layers can locate answer-related tokens.

As shown in Figure 3, GemFilter runs faster and consumes less GPU memory than SnapKV/H2O and standard attention (full KV cache) during the prompt computation phase. During the iterative generation phase, GemFilter has the same running time and GPU memory consumption as Snap-KV/H2O, both of which outperform standard attention. We discuss the complexity further in Section 3.2 theoretically and in Section 4.5 empirically. GemFilter significantly outperforms standard attention and SnapKV on the Needle in a Haystack benchmark (Section 4.1). Additionally, on Long-Bench, a multi-task benchmark designed to rigorously evaluate long-context understanding across

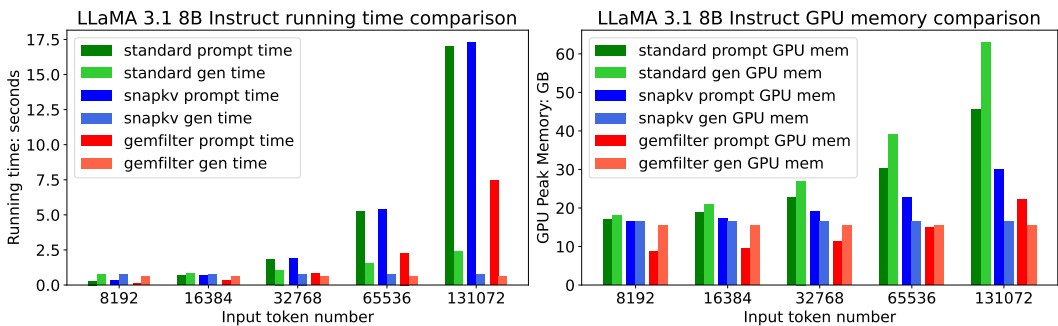

Figure 3: Comparison of time and GPU memory usage across different methods on LLaMA 3.1 8B Instruct. 'gemfilter' represents our method, using the 13th layer as the filter. It achieves a 2.4× speedup and reduces GPU memory usage by 30% compared to SnapKV. The iterative generation is evaluated on 50 tokens generation. Additional results can be found in Section 4.5.

various datasets, GemFilter achieves performance comparable to SnapKV/H2O (Section 4.2). Furthermore, our ablation study in Section 4.3 shows that our method is quite robust to the filter layer selection strategy and Section 4.4 shows that each component in our algorithm is essential.

**Our contributions and advantages are:**

- We found that LLMs can identify relevant tokens using attention matrices in the early layers, suggesting crucial information is recognized before the answer generation. Furthermore, LLMs explicitly summarize this information within specific filter layers. This observation provides insights into LLM mechanisms and opens avenues for LLM understanding and algorithm design.

- Leveraging this insight, we develop GemFilter, formulated in Algorithm 1, an inference strategy which utilizes early LLM layers as a filter to select and compress input tokens into a small subset to be processed by the full model (Figure 1). GemFilter achieves a 2.4× speedup and reduces GPU memory consumption by 30% compared to the state-of-the-art methods like SnapKV.

- GemFilter significantly outperforms both standard attention (all KV cache) and SnapKV on the Needle in a Haystack benchmark (Section 4.1), while maintaining performance comparable to SnapKV/H2O on the LongBench benchmark (Table 1).

- We provide a thorough ablation studies for the GemFilter in Section 4.3 and Section 4.4.

- Our approach offers several advantages: it is simple, training-free, and broadly applicable to various LLMs. Furthermore, it enhances interpretability by allowing humans to directly inspect the selected token sequence.

## 2 RELATED WORKS

**Generation Speed-up with Long Context Input.** One effective technique to accelerate autoregressive generation is KV cache compression/eviction. During generation, LLMs store the previous key and value matrices to reduce computational complexity. However, when the input context is long (e.g., 128K tokens), the memory consumption and running time associated with the KV cache dominate iterative generation. Many studies have focused on KV cache eviction. For instance, Ge et al. (2023) evict long-range contexts on attention heads to prioritize local contexts, using the KV cache only for heads that broadly attend to all tokens. Streaming LLM (Xiao et al., 2023) introduces an attention sink that retains only the first few tokens and the latest $k$ tokens in the KV cache to enable fast streaming generation. LOOK-M (Wan et al., 2024) applies KV eviction in the multi-modality so that the model only needs to look once for the image. LongWriter (Bai et al., 2024) uses KV eviction to enable LLMs to generate coherent outputs exceeding 20,000 words. MInference 1.0 (Jiang et al., 2024a) introduces ∧-shape, vertical-slash, and block-sparse attention head and determines the optimal KV cache pattern for each attention head offline and dynamically builds sparse indices based on the assigned query during inference. QuickLLaMA (Li et al., 2024a) classifies the KV cache to many subsets, e.g., query tokens, context tokens, global tokens, and local tokens, and only preserves some types of tokens in the KV cache. ThinK (Xu et al., 2024b) proposes a

query-dependent KV cache pruning method by pruning the least significant channel dimensions of the KV cache. H2O (Zhang et al., 2023) retains only tokens contributing to cumulative attention. SnapKV (Li et al., 2024b) evicts non-essential KV positions for each attention head based on observation windows. While the aforementioned studies focus on eviction and compression of the KV cache during the prompt computation phase to optimize the iterative generation phase, they do not reduce the running time or GPU memory usage during the prompt computation phase. In contrast, our method, GemFilter, achieves both reduced running time and GPU memory usage in the prompt computation phase, as well as during the iterative generation phase. We provide a more detailed comparison in Appendix B.

More related to our work, Li et al. (2023) compress input sequences by pruning redundancy in the context, making inputs more compact. However, they need to keep 50% of input tokens to keep the LLMs' performance, whereas GemFilter achieves comparable performance by only reserving 1% of input tokens. For further details, we refer the reader to Section 4.1. The LLMLingua series methods (Jiang et al., 2023b; Pan et al., 2024; Jiang et al., 2024b) present a coarse-to-fine approach for prompt compression. It leverages a budget controller to ensure semantic integrity even at high compression ratios, employs a token-level iterative compression algorithm to model interdependencies within the compressed content, and utilizes an instruction-tuning strategy to achieve distribution alignment across language models.

## 3 METHOD

**Notations and Preliminary.** While the Transformer and self-attention architecture (Vaswani et al., 2017) have already become overwhelmingly popular, we first introduce preliminary definitions to provide a better methodological connection to our proposed GemFilter method in Section 3.1.

For any positive integer $n$, we use $[n]$ to denote the set $\{1, 2, \cdots, n\}$. We use $\circ$ to denote function composition and $\odot$ to denote the Hardamard product. Let $n$ be the input token/prompt length, $d$ the hidden feature dimension, and $\mathcal{V}$ the vocabulary set. We now introduce the key concept of attention and transformers. We first define the query, key, and value matrices. It is important to note that during text generation, the key and value matrices are also referred to as the KV cache, as they are stored in GPU memory to reduce running time during the iterative prediction of the next token.

**Definition 3.1** (Single layer self-attention). *Let $Q \in \mathbb{R}^{n \times d}$ be the query matrix, $K \in \mathbb{R}^{n \times d}$ the key cache, and $V \in \mathbb{R}^{n \times d}$ the value cache. Let $M_c \in \{0,1\}^{n \times n}$ be the causal attention mask, where $(M_c)_{i,j}$ is 1 if $i \geq j$ and 0 otherwise. The self-attention function $\mathsf{Attn}$ is defined as:*

$$\mathsf{Attn}(Q, K, V) = M_c \odot \mathsf{Softmax}(QK^\top/\sqrt{d}) \cdot V$$

**Definition 3.2** (Multi-layer transformer). *Let $T \in \mathcal{V}^n$ represent the input tokens, and let $m$ denote the number of transformer layers. Let $g_i$ represent components in the $i$-th transformer layer other than self-attention, such as layer normalization, residual connections, and the MLP block, where $g_i : \mathbb{R}^{n \times d} \to \mathbb{R}^{n \times d}$ for any $i \in \{0, 1, \ldots, m\}$. Let $\mathsf{Attn}_i$ denote the self-attention module in the $i$-th transformer layer. We define an $m$-layer transformer $\mathsf{F}_{1:m} : \mathcal{V}^n \to \mathbb{R}^{n \times d}$ as*

$$\mathsf{F}_{1:m}(T) := g_m \circ \mathsf{Attn}_m \circ g_{m-1} \circ \cdots \circ g_1 \circ \mathsf{Attn}_1 \circ g_0 \circ \mathcal{E}(T) \quad \in \mathbb{R}^{n \times d},$$

*where $\mathcal{E}$ is the input embedding function mapping the input tokens to hidden features using the vocabulary dictionary, i.e., $\mathcal{E}(T) \in \mathbb{R}^{n \times d}$.*

Note that the above definitions use a single attention head for simplicity, but in practice, multi-head attention is used (Vaswani et al., 2017).

### 3.1 OUR ALGORITHM: GEMFILTER

We present our method, GemFilter, in Algorithm 1. We also present PyTorch code in Appendix D.1 for the reader's interests. The high-level idea is to run the LLM twice. In the first pass, we run only the early layers of the LLM to select the key input tokens. This corresponds to the prompt computation phase (Line 4-7 of Algorithm 1). This process selects the top $k$ tokens that receive the most attention from the last query token. In the second pass, we feed the selected tokens to the full LLM and run the generation function, corresponding to the iterative generation phase (Line 8). Below, we explain Algorithm 1 step by step.

---

**Algorithm 1** GemFilter: Generation with Token Selection Based on Early Layers

---

1: **procedure** SELECTIONGEN($\mathsf{F}_{1:m}, T \in [\mathcal{V}]^n, r \in [m], k \in [n]$)
2:              $\triangleright$ $\mathsf{F}_{1:m}$ : An $m$-layer transformer network; $T$: input sequence of tokens
3:          $\triangleright$ $r$: filter layer index for token selection; $k$: number of selected tokens
4:  Get $Q^{(r)}, K^{(r)}$ by doing a $r$-layer forward pass: $\mathsf{F}_{1:r}(T)$
5:              $\triangleright$ $Q^{(r)}, K^{(r)} \in \mathbb{R}^{n \times d}$: the $r$-th layer query, key
6:  $J \leftarrow \mathsf{topk\_index}(Q_n^{(r)} K^{(r)^\top}, k)$   $\triangleright$ $Q_n^{(r)}$: the last row of $Q^{(r)}$; $Q_n^{(r)} K^{(r)^\top} \in \mathbb{R}^n$ are attn scores
7:  Sort the indices in $J$            $\triangleright$ $J \subseteq [n]$ and $|J| = k$
8:  **return** GEN($\mathsf{F}_{1:m}, T_J$)    $\triangleright$ GEN is generation function, $T_J \in [\mathcal{V}]^k$ is a sub-sequence of $T$ on $J$
9: **end procedure**

---

The input of the algorithm is an $m$-layer transformer $\mathsf{F}_1$ (Definition 3.2), an input token sequence $T \in \mathcal{V}^n$, and two hyperparameters $r \leq m, k \leq n$, where $r$ represents the index of the filter layer for context token selection and $k$ denotes the number of tokens to select. For example, in the case of LLaMA 3.1 8B Instruct (Figure 1), we have $m = 32$, $r = 13$, and $k = 1024$.

In the first step (Line 4), we run only the first $r$ layers forward to serve as a filter, obtaining the $r$-th layer's query and key matrices, $Q^{(r)}$ and $K^{(r)}$. Note that we do not need to run all layers of the LLM on a long context input, thereby saving both computation time and memory (see detailed analysis in Section 3.2). In Line 6, we select token indices based on the $r$-th layer attention matrix. The selection is made by identifying the $k$ largest values from the last row of the attention matrix, i.e., the inner product between the last query token $Q_n^{(r)}$ and all key tokens $K^{(r)}$. For multi-head attention, the top-$k$ indices are selected based on the summation of the last row across the attention matrices of all heads. For instance, suppose we have $h$ attention heads, and let $Q^{(r,j)}, K^{(r,j)} \in \mathbb{R}^{n \times d}$ represent the query and key matrices for the $r$-th layer and $j$-th attention head. Then, we compute $J \leftarrow \mathsf{topk\_index}(\sum_{j=1}^h Q_n^{(r,j)} K^{(r,j)^\top}, k)$, where $J$ is a set of top $k$ index selection. Note that our method uses a single index set $J$, whereas SnapKV (Li et al., 2024b) and H2O (Zhang et al., 2023) use different index sets for each layer and attention head, resulting in $m \cdot h$ index sets in total. A detailed discussion is provided in Appendix B.

In Line 6, $J$ is sorted by inner product values. However, we need to re-sort $J$ so that the selected tokens follow their original input order, ensuring, for example, that the $\langle bos \rangle$ token is placed at the beginning. Line 7 performs this reordering operation. Finally, in Line 8, we can run any language generation function using the selected tokens $T_J$, which is a sub-sequence of $T$ on the index set $J$, across all layers. This generation is efficient as the input context length is reduced from $n$ to $k$, e.g., from 128K to 1024 tokens in Figure 1. Below, we provide a formal time complexity analysis.

## 3.2 RUNNING TIME AND MEMORY COMPLEXITY ANALYSIS

The results of our analysis on time complexity and GPU memory consumption are presented in Theorem 3.3 below, with the proof deferred to Appendix C.

**Theorem 3.3** (Complexity analysis). *Let $n$ be the input sequence (prompt) length and $d$ the hidden feature dimensions. In our Algorithm 1, GemFilter uses the $r$-th layer as a filter to select $k$ input tokens. Let SnapKV and H2O also use $k$ as their cache size. Assume the LLM has $m$ attention layers, each with $h$ attention heads, and each transformer layer's parameters consume $w$ GPU memory. Assuming that we generate $t$ tokens with the GEN function and $n \geq \max\{d, k, t\}$, the following table summarizes the complexity for standard attention, SnapKV and H2O, and GemFilter:*

| | Complexity | Standard attention | SnapKV and H2O | GemFilter |
|---|---|---|---|---|
| *Time* | *Prompt Comp.* | $\Theta(mhn^2d)$ | $\Theta(mhn^2d)$ | $\Theta(rhn^2d)$ |
| | *Iter. generation* | $\Theta(mh(nt + t^2)d)$ | $\Theta(mh(kt + t^2)d)$ | $\Theta(mh(k^2 + t^2)d)$ |
| *GPU mem.* | *Prompt Comp.* | $mw + 2mhnd$ | $mw + 2hnd + 2mhkd$ | $rw + 2hnd$ |
| | *Iter. generation* | $mw + 2mh(n+t)d$ | $mw + 2mh(k+t)d$ | $mw + 2mh(k+t)d$ |

Recall that there are two phases in text generation. The first phase is *prompt computation*, which involves attention computation on the long context input tokens and generating the KV cache. The second phase is *iterative generation*, where auto-regressive generation occurs based on the pre-computed KV cache. Theorem 3.3 demonstrates that GemFilter is faster and consumes less GPU memory than SnapKV/H2O and standard attention during the prompt computation phase. Additionally, during the iterative generation phase, GemFilter has the same running time and GPU memory consumption as SnapKV/H2O, which is significantly better than standard attention. This conclusion aligns with our experimental results in Section 4.5.

**Case Study.**  Let us consider the case $n \gg k \approx t$, e.g., $n =$128K, $k = t = 1024$ and $r < m$. During the prompt computation phase, we have the running time and the GPU memory consumption:

$$\text{Standard attention : SnapKV/H2O : GemFilter} = \Theta(m : m : r),$$
$$\text{Standard attention : SnapKV/H2O : GemFilter} \approx mw + mhnd : mw + hnd : rw + hnd,$$

We see that GemFilter has a lower time complexity and less GPU memory consumption than standard attention, SnapKV, and H2O. During the iterative generation phase, we have the running time and the GPU memory consumption:

$$\text{Standard attention : SnapKV/H2O : GemFilter} = \Theta(n : k : k),$$
$$\text{Standard attention : SnapKV/H2O : GemFilter} \approx w/hd + 2n : w/hd + 4k : w/hd + 4k,$$

As such, GemFilter has the same time complexity and GPU memory consumption as SnapKV/H2O, while significantly outperforming the standard attention. The running time bottleneck for all methods occurs during prompt computation, which takes $\Theta(mhn^2d)$ for standard attention, SnapKV, and H2O. In contrast, GemFilter only requires $\Theta(rhn^2d)$ for prompt computation, as it only processes the early layers of the LLMs to select and compress the input tokens during the first run. See detailed proof in Appendix C. Note that the GPU memory bottleneck for standard attention occurs during iterative generation, while for other methods, the memory bottleneck arises during prompt computation due to the reduced KV cache. GemFilter consumes less GPU memory than SnapKV and H2O because it only requires loading some layer model weights when processing the long context input in its first run. Our empirical results in Section 4.5 support our complexity analysis findings.

## 4 EXPERIMENTS

**Model and Datasets.**  We evaluated our approach using three popular long-context models: LLaMA 3.1 8B Instruct[1] (Dubey et al., 2024), Mistral Nemo 12B Instruct[2] (Jiang et al., 2023a), and Phi 3.5 Mini 3.8B Instruct[3] (Abdin et al., 2024), all of which support an input token length of 128K. We compared our method, GemFilter, against standard attention and two state-of-the-art methods, SnapKV (Li et al., 2024b) and H2O (Zhang et al., 2023)[4]. For our experiments, we used two popular datasets: Needle in a Haystack (Kamradt, 2024) (Section 4.1) and LongBench (Bai et al., 2023) (Section 4.2). More implementation details are provided in Appendix D.2.

**Filter Layer.**  Except for Section 4.3, for context selection, we always use the index of 13 out of 32, 19 out of 40, and 19 out of 32 layers as the input filter for LLaMA 3.1, Mistral Nemo and Phi 3.5, respectively. In Section 4.3, we provide an ablation study for the filter layer choice.

### 4.1 NEEDLE IN A HAYSTACK

The Needle in a Haystack (Kamradt, 2024) benchmark serves as a pressure test, challenging LLMs to retrieve accurate information from a specific sentence (the 'needle') hidden within an extensive document (the 'haystack'), where the sentence can appear at any arbitrary location. The difficulty increases as the length of the haystack grows. We use input lengths of 60K for Mistral Nemo 12B

---

[1] https://huggingface.co/meta-llama/Meta-Llama-3.1-8B-Instruct

[2] https://huggingface.co/mistralai/Mistral-Nemo-Base-2407

[3] https://huggingface.co/microsoft/Phi-3.5-mini-instruct

[4] While there are many other generation acceleration methods, they may not be directly comparable to ours as they use orthogonal techniques. We refer the reader to Section 2 for further details.

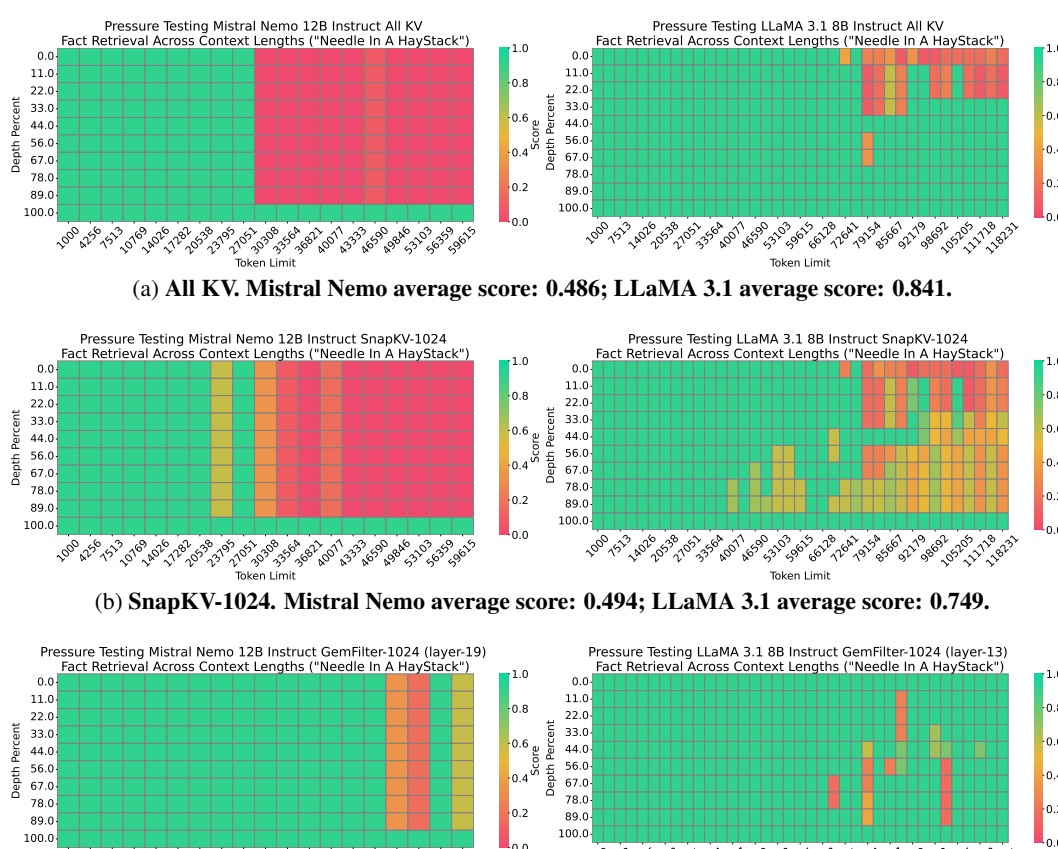

(a) **All KV. Mistral Nemo average score: 0.486; LLaMA 3.1 average score: 0.841.**

(b) **SnapKV-1024. Mistral Nemo average score: 0.494; LLaMA 3.1 average score: 0.749.**

(c) **GemFilter-1024. Mistral Nemo average score: 0.838; LLaMA 3.1 average score: 0.887.**

Figure 4: Needle in a Haystack performance comparison of different methods using the Mistral Nemo 12B Instruct model (left column) and the LLaMA 3.1 8B Instruct model (right column). Results for the Phi 3.5 Mini 3.8B Instruct model are provided in Appendix D.3. The $x$-axis represents the length of the input tokens, while the $y$-axis shows the position depth percentage of the 'needle' information (e.g., 0% indicates the beginning, and 100% indicates the end). A higher score reflects better performance, meaning more effective retrieval of the 'needle' information. GemFilter significantly outperforms both standard attention (full KV cache) and SnapKV.

Instruct and 120K for LLaMA 3.1 8B Instruct, as these are the maximum lengths for standard attention on two A100-40GB GPUs. The KV cache size is set to 1024 for both SnapKV and GemFilter. In Figure 4, we see that GemFilter significantly outperforms both All KV (standard attention) and SnapKV with Mistral Nemo and LLaMA 3.1.[5] The Needle in a Haystack results suggest that our method, GemFilter, achieves superior retrieval performance for long input contexts compared to SnapKV and standard attention. Additional results are provided in Appendix D.3.

## 4.2 LONGBENCH

LongBench (Bai et al., 2023) is a multi-task benchmark designed to rigorously evaluate long-context understanding capabilities across various datasets, including single- and multi-document Question Answering (QA), summarization, few-shot learning, and synthetic tasks. We evaluate the English-only dataset, following Li et al. (2024b); Xu et al. (2024b). Note that we do not use a chat template in Table 1. See Table 3 in Appendix D.7 for more results of using a chat template.

---

[5]H2O cannot be implemented with FlashAttention due to its cumulative attention score strategy and is therefore unable to handle super long input contexts, which is why we exclude it here, following Li et al. (2024b); Xu et al. (2024b).

Table 1: Performance comparison on LongBench across various LLMs and methods. A larger number means better performance. The best score is **boldfaced**.

| Method | Single-Document QA | | | Multi-Document QA | | | Summarization | | | Few-shot Learning | | | Synthetic | | Average |
| --- | --- | --- | --- | --- | --- | --- | --- | --- | --- | --- | --- | --- | --- | --- | --- |
| | NrtvQA | Qasper | MF-en | HotpotQA | 2WikiMQA | Musique | GovReport | QMSum | MultiNews | TREC | TriviaQA | SAMSum | PCount | PRe | |
| **LLaMA 3.1 8B Instruct** | | | | | | | | | | | | | | | |
| All KV | 32.02 | 13.04 | 27.34 | 16.23 | 16.05 | 11.22 | 34.52 | 23.41 | **26.89** | **73.0** | 91.64 | 43.8 | 7.16 | **97.73** | **36.72** |
| H2O-4096 | 22.94 | 12.61 | 26.48 | 16.63 | 15.81 | 10.14 | 33.51 | 23.47 | 26.81 | 69.0 | 91.15 | **43.97** | 6.66 | 71.67 | 33.63 |
| MInference | 27.52 | **14.72** | 28.89 | 17.55 | 15.22 | 10.58 | **34.76** | 22.34 | 26.64 | 72.5 | 89.78 | 41.94 | 7.59 | 92.91 | 35.92 |
| LLMLingua-1024 | 11.73 | 6.28 | 12.43 | 13.82 | 12.92 | 8.15 | 22.82 | 20.18 | 23.32 | 24.0 | 66.75 | 24.02 | **9.09** | 4.24 | 18.55 |
| SnapKV-1024 | 31.98 | 11.17 | 25.33 | 14.81 | 15.73 | 10.69 | 26.95 | 22.89 | 25.86 | 67.5 | 91.89 | 42.85 | 7.67 | 98.16 | 35.25 |
| GemFilter-1024 | 20.71 | 11.0 | **29.28** | 19.12 | 17.01 | 13.01 | 30.37 | 21.75 | 25.17 | 63.0 | 90.7 | 42.5 | 7.15 | 92.22 | 34.50 |
| SnapKV-2048 | 31.45 | 11.94 | 26.24 | 15.73 | 16.03 | 11.66 | 29.64 | 23.24 | 26.44 | 69.5 | 91.48 | 42.68 | 7.21 | 98.03 | 35.80 |
| GemFilter-2048 | 24.36 | 12.63 | 25.39 | **19.58** | **17.03** | **14.11** | 33.15 | 22.31 | 26.49 | 69.5 | 91.59 | 42.64 | 4.61 | **98.75** | 35.87 |
| SnapKV-4096 | **32.13** | 13.12 | 27.38 | 16.11 | 16.08 | 11.6 | 32.39 | 23.47 | 26.76 | 71.5 | 91.64 | 43.46 | 7.33 | 97.24 | 36.44 |
| GemFilter-4096 | 25.66 | 12.95 | 27.38 | 17.76 | 15.6 | 12.02 | 34.17 | 23.25 | 26.87 | 70.0 | **92.36** | 43.34 | 5.96 | 98.0 | 36.09 |
| **Mistral Nemo 12B Instruct** | | | | | | | | | | | | | | | |
| All KV | 28.91 | 40.74 | 54.65 | 52.15 | 48.36 | 30.28 | **30.66** | **23.53** | 26.31 | 75.0 | 89.66 | 44.32 | 4.5 | 100.0 | 46.36 |
| H2O-4096 | **31.61** | 39.52 | 54.75 | 47.83 | 48.09 | 27.0 | 30.44 | 23.21 | 26.42 | 72.5 | 89.76 | 44.47 | 3.0 | 73.0 | 43.69 |
| LLMLingua-1024 | 19.24 | 16.92 | 21.43 | 30.94 | 25.09 | 13.24 | 21.96 | 19.8 | 23.94 | 24.5 | 68.48 | 33.33 | 4.0 | 5.0 | 23.42 |
| SnapKV-1024 | 26.42 | 38.49 | 52.96 | 51.21 | 47.86 | 27.06 | 24.32 | 22.66 | 25.52 | 73.0 | 89.82 | 43.16 | 3.5 | 100.0 | 44.71 |
| GemFilter-1024 | 27.53 | 40.68 | 53.86 | 55.51 | **55.43** | 34.11 | 27.25 | 21.16 | 25.56 | 69.0 | 87.32 | 42.49 | 4.0 | 88.06 | 45.14 |
| SnapKV-2048 | 25.85 | 40.69 | 54.48 | 51.96 | 49.06 | 26.95 | 26.29 | 23.17 | 25.9 | 74.5 | 89.66 | 43.89 | 4.0 | 99.5 | 45.42 |
| GemFilter-2048 | 29.27 | **41.53** | **54.91** | 57.62 | 54.97 | 35.09 | 29.34 | 22.58 | 26.19 | 72.0 | 89.65 | **44.93** | 4.0 | 97.5 | **47.11** |
| SnapKV-4096 | 27.92 | 40.9 | 54.75 | 51.69 | 48.16 | 29.19 | 29.17 | 23.36 | 26.35 | 75.0 | 89.66 | 43.93 | 4.5 | 100.0 | 46.04 |
| GemFilter-4096 | 30.29 | 39.9 | 56.48 | **58.78** | 51.48 | 32.81 | 30.32 | 23.21 | **26.48** | 71.5 | 90.24 | 42.13 | 2.0 | 99.5 | 46.79 |
| **Phi 3.5 Mini 3.8B Instruct** | | | | | | | | | | | | | | | |
| All KV | **27.51** | 17.23 | 35.63 | 21.7 | 25.7 | 11.68 | 34.14 | **23.17** | 24.95 | **71.5** | 87.37 | 13.08 | **7.17** | 83.85 | **34.62** |
| H2O-4096 | 19.74 | 16.23 | 34.17 | 21.02 | 23.05 | 11.09 | 33.42 | 21.95 | 24.95 | 67.5 | 86.13 | 16.71 | 1.55 | 47.46 | 30.31 |
| LLMLingua-1024 | 8.58 | 6.74 | 14.93 | 12.37 | 11.01 | 4.48 | 21.23 | 17.08 | 20.75 | 24.0 | 56.09 | 23.01 | 0.96 | 3.79 | 16.07 |
| SnapKV-1024 | 24.31 | 16.03 | 34.93 | 20.72 | 26.02 | 13.74 | 28.27 | 22.03 | 24.02 | 67.5 | **87.71** | 14.57 | 6.08 | **85.6** | 33.68 |
| GemFilter-1024 | 16.57 | 18.29 | 35.91 | 24.22 | 26.1 | 9.7 | 30.29 | 18.96 | 23.64 | 64.5 | 85.85 | **23.02** | 0.2 | 81.12 | 32.74 |
| SnapKV-2048 | 26.41 | 16.59 | **36.99** | 21.8 | 26.07 | 12.57 | 30.88 | 22.37 | 24.51 | 69.5 | 87.54 | 13.13 | 6.57 | 83.92 | 34.20 |
| GemFilter-2048 | 19.63 | 14.84 | 35.99 | 21.38 | 19.72 | 10.13 | 32.39 | 21.24 | 24.71 | 65.0 | 86.49 | 20.47 | 2.17 | 69.5 | 31.69 |
| SnapKV-4096 | 27.25 | 17.42 | 36.9 | 21.37 | 25.42 | 12.55 | 32.9 | 22.6 | 24.87 | 70.5 | 87.45 | 13.28 | 6.81 | 84.04 | 34.53 |
| GemFilter-4096 | 20.95 | **19.98** | 35.22 | **28.82** | **28.21** | **13.98** | **34.2** | 22.45 | **25.08** | 64.5 | 85.86 | 18.68 | 3.43 | 65.56 | 33.35 |

For each LLM, we evaluate GemFilter and SnapKV with selected tokens/KV caches of 1024, 2048, and 4096. We also evaluated standard attention (all KV cache) and H2O with a KV cache size of 4096 on the LongBench dataset to further demonstrate the performance of GemFilter, following Li et al. (2024b). Table 1 shows a negligible performance drop in LLMs using GemFilter compared to standard attention, even with only 1024 selected tokens. In some cases, GemFilter even outperforms standard attention, such as GemFilter-2048 for Mistral Nemo 12B Instruct. It demonstrates significantly better performance than H2O and comparable performance with SnapKV. Furthermore, GemFilter effectively filters key information in long contexts, provides interpretable summaries, and compresses the input context effectively, e.g., it reduces input tokens to an average of 8% when using 1024 tokens, and 32% when using 4096, with negligible accuracy drops.

In the section, we also evaluated on two important baselines, MInference (Jiang et al., 2024a) and LLMLingua (Jiang et al., 2023b)[6]. We can see that MInference (Jiang et al., 2024a) has compatible performance with SnapKV, while it requires offline to determine the best attention pattern, which cannot save the prompt computation phase running time. We can see that although LLMLingua (Jiang et al., 2023b) achieves a good comparison rate, the performance may not be satisfactory.

## 4.3 ABLATION STUDY: FILTER LAYER CHOICE

In this section, we explore which layer should be chosen as the input filter. First, we aim to determine which layer of the LLM can best identify the position of the needle information. In Figure 5, we

---

[6]We skip LongLLMLingua Jiang et al. (2024b) for a fair comparison, as it requires explicitly separating the input context into text information and questions, while other methods do not require that.

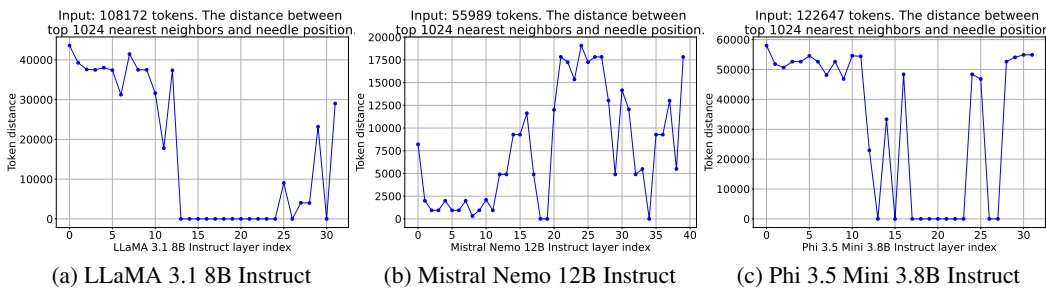

(a) LLaMA 3.1 8B Instruct          (b) Mistral Nemo 12B Instruct          (c) Phi 3.5 Mini 3.8B Instruct

Figure 5: Distance between the needle position and selected token index position across three LLMs. The position depth percentage of the "needle" information is 50%. The $x$-axis means the layer index of different LLMs. The $y$-axis means $\min(\text{topk\_index} - \text{niddle\_index})$. When $y = 0$, it means the needle information is covered by the selected token. The needle information has been successfully discovered in the early layers of all three LLMs.

plot the distance between the needle's position and the selected token index across all layers in the LLM. The results reveal three stages in the prompt computation of LLMs. In the first stage, the initial layers preprocess the input context and search for the 'needle'. In the second stage, some early to middle layers identify the needle information. Finally, in the third stage, the LLM prepares to generate the output based on the selected tokens.

Table 2: Performance of our method on LongBench using different layers as an input filter. A larger number means better performance. The best score is **boldfaced**.

| Filter layer | Single-Document QA | | | Multi-Document QA | | | Summarization | | | Few-shot Learning | | | Synthetic | | Average |
| | NrtvQA | Qasper | MF-en | HotpotQA | 2WikiMQA | Musique | GovReport | QMSum | MultiNews | TREC | TriviaQA | SAMSum | PCount | PRe | |
| --- | --- | --- | --- | --- | --- | --- | --- | --- | --- | --- | --- | --- | --- | --- | --- |
| **LLaMA 3.1 8B Instruct (32 layers)** | | | | | | | | | | | | | | | |
| layer-1 | 16.32 | 7.38 | 13.86 | 13.9 | 13.21 | 5.22 | 25.61 | 20.09 | 24.51 | 47.0 | 76.59 | 39.78 | 2.55 | 23.01 | 23.50 |
| layer-7 | 16.89 | 6.83 | 13.47 | 13.78 | 12.23 | 9.67 | 26.56 | 19.49 | 24.55 | 58.0 | 84.87 | 41.07 | 6.5 | 50.69 | 27.47 |
| layer-12 | 15.53 | 7.73 | 16.53 | 17.08 | 13.33 | 9.88 | 28.94 | 20.32 | 25.01 | 58.0 | 88.16 | 40.42 | 8.36 | 43.06 | 28.03 |
| layer-13 | 20.71 | 11.0 | **29.28** | 19.12 | 17.01 | 13.01 | **30.37** | 21.75 | 25.17 | **63.0** | **90.7** | **42.5** | 7.15 | 92.22 | **34.50** |
| layer-14 | 21.14 | **13.06** | 25.45 | 20.89 | **17.32** | 12.9 | 29.85 | **22.06** | 24.91 | 62.0 | 89.88 | 42.33 | 6.17 | 92.17 | 34.30 |
| layer-19 | 19.06 | 11.69 | 27.12 | **20.98** | 16.98 | **14.04** | 29.17 | 21.88 | **25.18** | 58.0 | 89.65 | 40.4 | **8.75** | **94.84** | 34.12 |
| layer-25 | **24.74** | 12.33 | 26.18 | 18.56 | 16.3 | 12.54 | 28.66 | 21.75 | 25.14 | 61.5 | 88.78 | 39.47 | 8.67 | 90.59 | 33.94 |
| layer-31 | 20.62 | 9.13 | 17.51 | 19.13 | 13.76 | 10.07 | 28.21 | 21.11 | 25.16 | 58.0 | 88.4 | 42.37 | 8.23 | 58.8 | 30.04 |

We then use the first layer that accurately identifies the needle's position as the input filter. In our experiments, we find that this layer remains consistent across different inputs. As shown in Table 2, performance first increases and then decreases as we select the input filter layer from the beginning to the end. The peak performance is observed at the 13th layer, which supports our layer selection strategy. Performance remains robust between layers 13 and 25, providing flexibility in layer selection. Exploring the distinct functions of different layers presents an interesting direction for future research.

## 4.4 MORE ABLATION STUDY

To understand the intuition behind selecting tokens with the most attention specifically from the last query, we study using different rows rather than the last row in the attention matrix for indices selection, as shown in Figure 2 in Appendix D.4. In Figure 9, we introduce two methods: (a) selecting middle rows of the attention matrix and (2) selecting rows with the largest $\ell_2$ norm. Both methods fail in the Needle in a Haystack task, verifying that selecting the last query token is essential.

Note that the performance improvement of GemFilter may stem from two factors: (1) the selection of important tokens, and (2) the re-computation of these tokens, which might mitigate issues like "lost-in-the-middle". To understand whether both factors made contributions, we provide an ablation study to isolate the contribution of each factor in Figure 10 of Appendix D.5. Furthermore, in Appendix D.6 Figure 11, we show the index selection difference between Gemfilter and SnapKV.

### 4.5 RUNNING TIME AND GPU MEMORY CONSUMPTION

In this section, we compare the running time and GPU memory consumption of different methods with FlashAttention (Dao et al., 2022; Dao, 2023; Shah et al., 2024) support.[7] The iterative generation running time and memory consumption are evaluated on 50 tokens generation. As shown in Figure 3, our method, GemFilter, achieves a 2.4× speedup compared to SnapKV and standard attention, with 30% and 70% reductions in GPU memory usage, respectively. It saves both running time and GPU memory by processing the long input context only during the first stage, as described in Section 4.3. For the latter two stages, the LLMs only need to handle compressed inputs. In Figure 6, we present a comparison of running time and GPU memory consumption for Mistral Nemo 12B Instruct and Phi 3.5 Mini 3.8B Instruct using various methods. GemFilter runs faster and uses less GPU memory than the state-of-the-art methods, as discussed above. Additionally, Figure 3 and Figure 6 further support our Theorem 3.3 in Section 3.2.

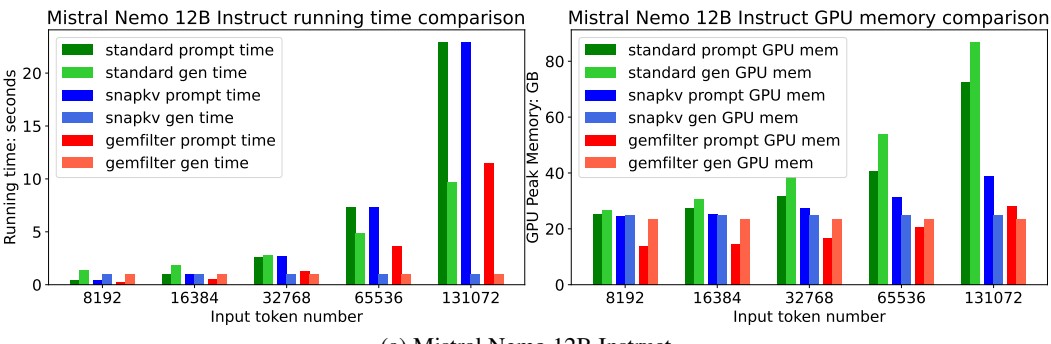

(a) Mistral Nemo 12B Instruct

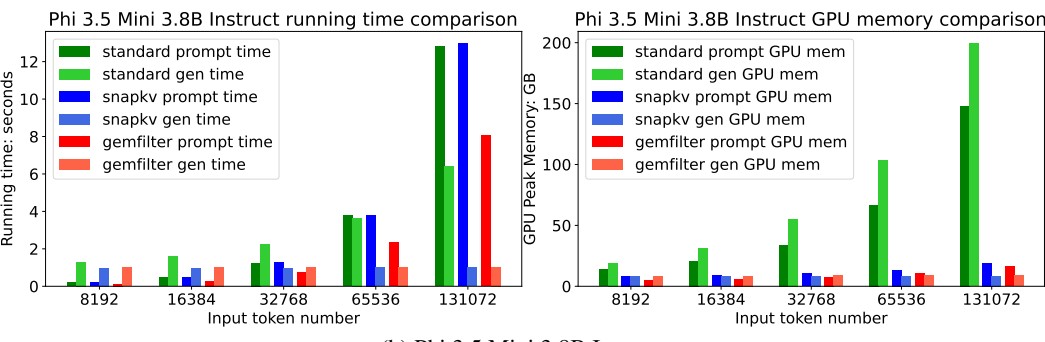

(b) Phi 3.5 Mini 3.8B Instruct

Figure 6: Comparison of time and GPU memory usage across different methods on Mistral Nemo 12B Instruct and Phi 3.5 Mini 3.8B Instruct. GemFilter uses the 19th layer as an input filter for both LLMs. It achieves a 2.4× speedup and reduces GPU memory usage by 30% compared to SnapKV.

## 5 CONCLUSION

In this work, we presented a novel approach, GemFilter, to accelerate LLM inference and reduce memory consumption for long context inputs. By leveraging the ability of early LLM layers to identify relevant information, GemFilter achieves significant improvements over existing techniques. It demonstrates a 2.4× speedup and 30% reduction in GPU memory usage compared to SOTA methods, while also showing superior performance on the Needle in a Haystack benchmark. Our approach is simple, training-free, applicable to various LLMs, and offers enhanced interpretability by directly inspecting selected tokens. These results not only provide practical benefits for LLM deployment, but also provide insight into a better understanding of LLM internal mechanisms.

---

[7]We exclude H2O as it does not support FlashAttention and thus requires more GPU memory and running time than standard attention during prompt computation.

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

# Appendix

## A    MORE PRELIMINARY

In this section, we introduce some key definitions of language modeling modules. We begin with the input embedding function and the output embedding function. They are functions that bridge between the input token space and the real vector space.

**Definition A.1** (Input embedding function and input tokens). *The input embedding function $\mathcal{E}$ : $\mathcal{V}^n \to \mathbb{R}^{n \times d}$ maps the input tokens to hidden features using the vocabulary dictionary $D^{\mathrm{voc}} \in \mathbb{R}^{|\mathcal{V}| \times d}$. Let $T \in \mathcal{V}^n$ be input tokens. Then, we have $\mathcal{E}(T) \in \mathbb{R}^{n \times d}$ and $\mathcal{E}(T)_i = D^{\mathrm{voc}}_{T_i} \in \mathbb{R}^d$ for any $i \in [n]$.*

**Definition A.2** (Output embedding function). *The output embedding function $\mathcal{G} : \mathbb{R}^d \to \mathbb{R}^{|\mathcal{V}|}$ maps hidden features to the probability logits of the vocabulary dictionary.*

We introduce Softmax, which allows self-attention to learn the probability distribution rather than function anymore.

**Definition A.3** (Softmax). *Let $z \in \mathbb{R}^n$. We define $\mathsf{Softmax} : \mathbb{R}^n \to \mathbb{R}^n$ satisfying*

$$\mathsf{Softmax}(z) := \exp(z)/\langle \exp(z), \mathbf{1}_n \rangle.$$

## B    DETAILED COMPARISON WITH OTHER METHODS

GemFilter reduces both running time and GPU memory usage in both the prompt computation and iterative generation phases, whereas SnapKV (Li et al., 2024b) and H2O (Zhang et al., 2023) focus only on the iterative generation phase. During the prompt computation phase, standard attention computes and stores the entire KV cache for all layers in GPU memory, which is used during the generation phase. SnapKV and H2O, on the other hand, compute the entire KV cache for all layers but only store a portion of it in GPU memory (e.g., $k = 1024$). They use the selected KV cache for memory-efficient generation. SnapKV selects important clustered positions of the KV cache from an 'observation' window located at the end of the prompt, while H2O greedily drops tokens based on cumulative attention scores to retain only a small portion of the KV cache. In contrast, GemFilter avoids computing the KV cache for all layers during the prompt computation phase.

Compared to SnapKV and H2O, there are two additional differences. First, SnapKV and H2O maintain separate index sets for each layer and attention head, resulting in $m \cdot h$ index sets in total. This leads to different behaviors across attention heads, making their intermediate mechanisms more difficult to interpret. On the other hand, GemFilter uses a single index set, $J$, allowing for easier interpretability by enabling the printing of the selected sequence for human review before the second run (see a real example in Figure 1). Another distinction lies in how positional embeddings are handled. In SnapKV and H2O, the maximum positional embedding distance is $n + t$, as the same positional embedding is used in both the prompt computation and iterative generation phases. However, in GemFilter's second run, the maximum positional embedding distance is reduced to $k + t$ because the input token length is reduced from $n$ to $k$, and the RoPE function[8] is re-computed. This reduction makes GemFilter more efficient, as the model can better handle shorter input sequences, as demonstrated in Figure 4 (a).

## C    PROOF OF TIME COMPLEXITY

**Theorem C.1** (Complexity analysis. Restatement of Theorem 3.3). *Let $n$ be the input sequence (prompt) length and $d$ the hidden feature dimensions. In our Algorithm 1, GemFilter uses the $r$-th layer as a filter to select $k$ input tokens. Let SnapKV and H2O also use $k$ as their cache size. Assume*

---

[8]RoPE is the rotary positional embedding (Su et al., 2024), encoding the positional information of tokens.

*the LLM has $m$ attention layers, each with $h$ attention heads, and each transformer layer's parameters consume $w$ GPU memory. Assuming that we generate $t$ tokens with the GEN function and $n \geq \max\{d, k, t\}$, the following table summarizes the complexity for standard attention, SnapKV and H2O, and GemFilter:*

| | *Complexity* | *Standard attention* | *SnapKV and H2O* | *GemFilter* |
|---|---|---|---|---|
| *Time* | *Prompt Comp.* | $\Theta(mhn^2d)$ | $\Theta(mhn^2d)$ | $\Theta(rhn^2d)$ |
| | *Iter. generation* | $\Theta(mh(nt + t^2)d)$ | $\Theta(mh(kt + t^2)d)$ | $\Theta(mh(k^2 + t^2)d)$ |
| *GPU mem.* | *Prompt Comp.* | $mw + 2mhnd$ | $mw + 2hnd + 2mhkd$ | $rw + 2hnd$ |
| | *Iter. generation* | $mw + 2mh(n + t)d$ | $mw + 2mh(k + t)d$ | $mw + 2mh(k + t)d$ |

*Proof of Theorem 3.3.* We prove each method separately.

**Proof of standard attention:**

During prompting computation, it takes $\Theta(mhn^2d)$ time complexity, as there are $m$ transformer layers, each layer has $h$ attention head, and each head takes $\Theta(n^2d)$ to calculate the attention ($\text{Attn}_i$ in Definition 3.2) and $\Theta(nd)$ for other operations ($g_i$ in Definition 3.2).

During iterative generation, it takes $\Theta(mh(nt + t^2)d)$ time complexity.

During prompting computation, $mw$ GPU memory consumption is taken for the model weights and $2mhnd$ GPU memory consumption for the KV cache.

During iterative generation, it takes $mw$ GPU memory consumption for the model weights and $2mh(n + t)d$ GPU memory consumption for the KV cache. **Proof of SnapKV and H2O:**

During prompting computation, it takes $\Theta(mhn^2d)$ time complexity, which is the same as standard attention.

During iterative generation, it takes $\Theta(mh(kt + t^2)d)$ time complexity, as it reduces the KV cache size from $n$ to $k$.

During prompting computation, $mw$ GPU memory is consumed for the model weights, $2hnd$ for the selection of the key-value matrix for each layer, and $2mhkd$ for the selected KV cache.

During iterative generation, $mw$ GPU memory is consumed for the model weights and $2mh(k + t)d$ GPU memory is consumed for the KV cache.

**Proof of our Algorithm 1 GemFilter:**

During prompting computation, GemFilter takes $\Theta(rhn^2d)$ time complexity, which is faster than other methods.

During iterative generation, it takes $\Theta(mh(k^2 + kt + t^2)d) = \Theta(mh(k^2 + t^2)d)$ time complexity, as it reduces the KV cache size from $n$ to $k$.

During prompting computation, $rw + 2hnd$ GPU memory is consumed for the model weights and the selection of the key value matrix for each layer.

During iterative generation, $mw + 2mh(k + t)d$ GPU memory is consumed for the KV cache and model weights.

Thus, we finish the proof. □

# D MORE DETAILS ABOUT EXPERIMENTS

## D.1 PYTORCH CODE

We provide the PyTorch code of Algorithm 1 GemFilter below, where our method only needs a few lines of adaptation based on standard attention[9].

---

[9]https://github.com/huggingface/transformers/blob/v4.43-release/src/transformers/models/mistral/modeling_mistral.py

```
1  # find the selected input for the specific attention layer
2  def find_context(self, query_states, key_states, k):
3      # repeat kv for group query attention
4      key_states = repeat_kv(key_states, self.num_key_value_groups)
5      # only use the last query token for the top k selection
6      top_k_indices = top_index(key_states, query_states[:, :, -1:, :], k)
7      # sort the index into the correct order
8      return torch.sort(top_k_indices, dim=-1).indecies
9
10 def top_index(keys, queries, k, kernel=5):
11     # calculate the inner product
12     in_pro = torch.matmul(queries, keys.transpose(-1, -2))
13     # cumulate the score over all attention heads in one attention layer
14     in_pro = torch.sum(in_pro, dim=1, keepdim=True)
15     # use 1D pooling for clustering, similar as SnapKV
16     in_pro = F.avg_pool1d(in_pro, kernel=kernel, padding=kernel//2,
       stride=1)
17     return torch.topk(in_pro, k, dim=-1).indices
```

## D.2 IMPLEMENTATION DETAILS

All the Needle in a Haystack and LongBench experiments run on A100-40GB GPUs. All the experiments of running time and memory complexity are evaluated on H100-80GB GPUs. We use HuggingFace v4.43 PyTorch implementation. There is no randomness or training in all baseline methods or our method. For the SnapKV/H2O, we use 32 recent size/observation window, which is the optimal choice suggested by Li et al. (2024b); Xu et al. (2024b). However, GemFilter does not have an observation window. We use a maximum pooling kernel size (line 16 of the PyTorch code below) of 5 for SnapKV and our method. For generation, we use standard generation (greedy generation)[10], where *num_beams=1, do_sample = False*.

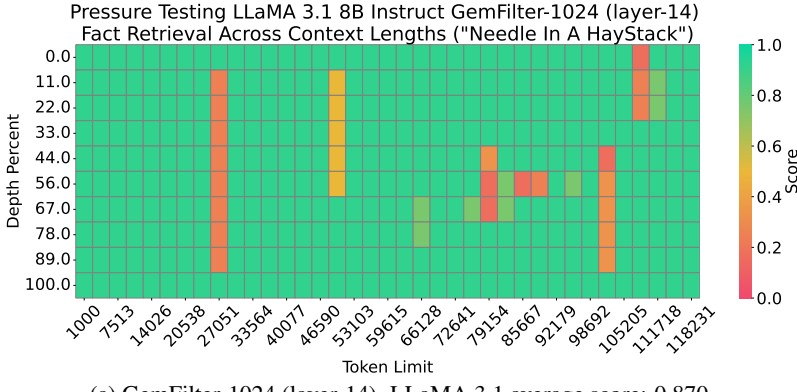

(a) GemFilter-1024 (layer-14). LLaMA 3.1 average score: 0.870.

Figure 7: Needle in a Haystack performance comparison of different filter layers with LLaMA 3.1 8B Instruct model. The $x$-axis represents the length of the input tokens, while the $y$-axis shows the position depth percentage of the 'needle' information (e.g., 0% indicates the beginning, and 100% indicates the end). A higher score reflects better performance, meaning more effective retrieval of the 'needle' information.

## D.3 MORE NEEDLE IN A HAYSTACK

We provide more results of Section 4.1 here. In Figure 8, GemFilter outperforms All KV (standard attention) and SnapKV by a large margin with Phi 3.5 Mini 3.8B Instruct. In Figure 7, we use layer 14 of LLama 3.1 as the input filter layer, which is an empirical support of the ablation study in Section 4.3, as it can also obtain good performance on the Needle in a Haystack benchmark.

---

[10]https://huggingface.co/docs/transformers/v4.43.2/en/main_classes/text_generation

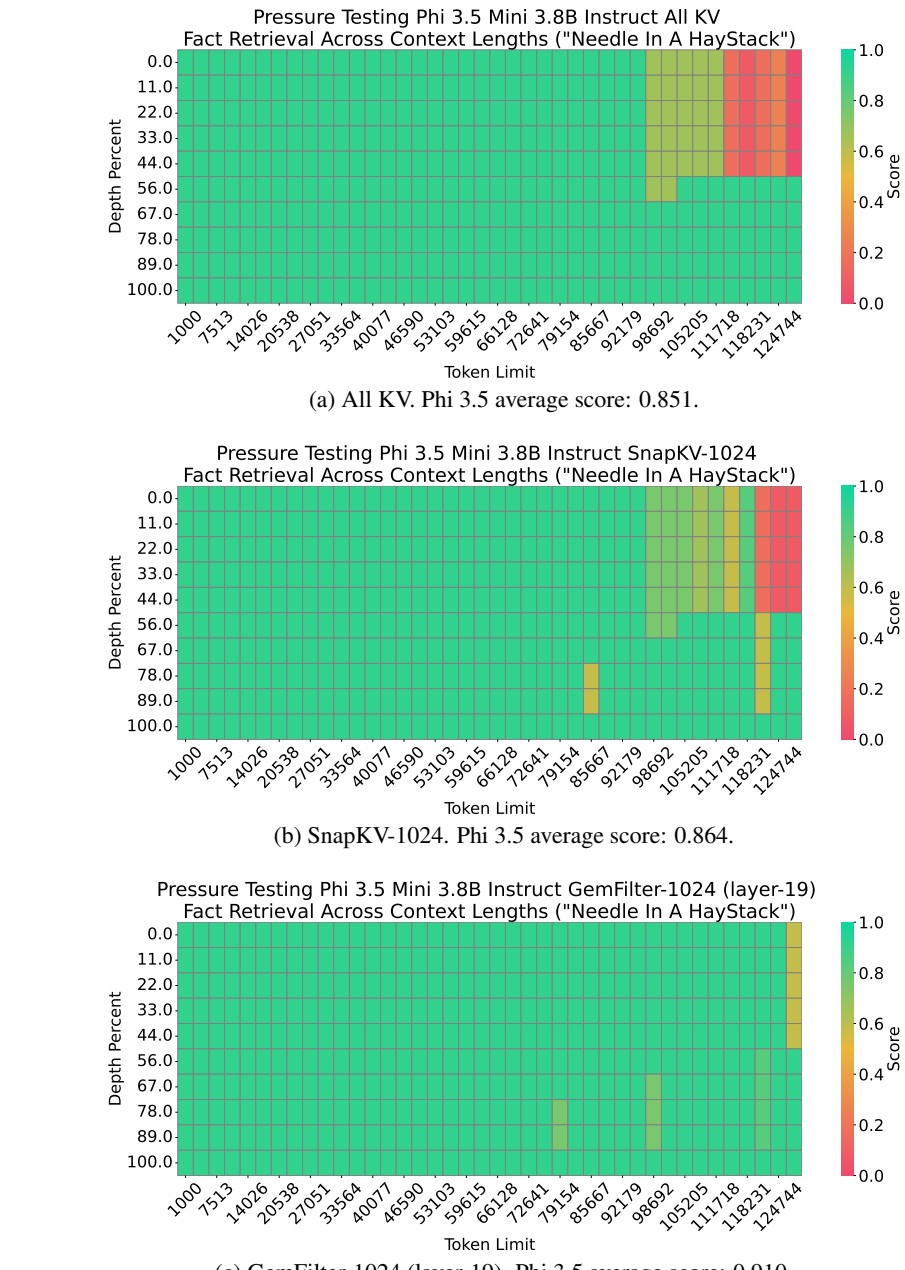

(a) All KV. Phi 3.5 average score: 0.851.

(b) SnapKV-1024. Phi 3.5 average score: 0.864.

(c) GemFilter-1024 (layer-19). Phi 3.5 average score: 0.910.

Figure 8: Needle in a Haystack performance comparison of different methods using the Phi 3.5 Mini 3.8B Instruct model. The $x$-axis represents the length of the input tokens, while the $y$-axis shows the position depth percentage of the 'needle' information (e.g., 0% indicates the beginning, and 100% indicates the end). A higher score reflects better performance, meaning more effective retrieval of the 'needle' information. GemFilter significantly outperforms both standard attention (full KV cache) and SnapKV.

## D.4 ABLATION STUDY ON ROW SELECTION

To understand the intuition behind selecting tokens with the most attention specifically from the last query, we study using different rows rather than the last row in the attention matrix for indices selection, as shown in Figure 2.

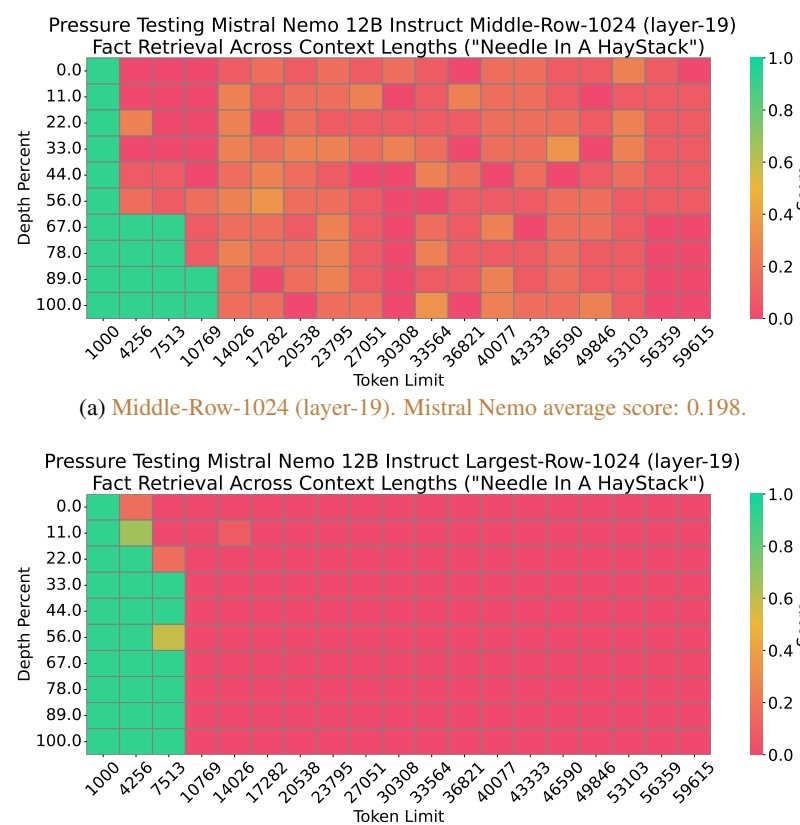

(a) Middle-Row-1024 (layer-19). Mistral Nemo average score: 0.198.

(b) Largest-Row-1024 (layer-19). Mistral Nemo average score: 0.125.

Figure 9: Needle in a Haystack performance comparison of different methods using the Mistral Nemo 12B Instruct model. The $x$-axis represents the length of the input tokens, while the $y$-axis shows the position depth percentage of the 'needle' information (e.g., 0% indicates the beginning, and 100% indicates the end). A higher score reflects better performance, meaning more effective retrieval of the 'needle' information. (a) is using the middle row to select top $k$ indices and (b) is using the row with largest $\ell_2$ norm to select top $k$ indices.

In Figure 9, we introduce two methods: (a) selecting the middle rows of the attention matrix and (2) selecting rows with the largest $\ell_2$ norm. As we can see, both methods fail in the Needle in a Haystack task. It shows that selecting the last query token is essential in our method.

## D.5 ABLATION STUDY ON RUNS

Note that the performance improvement of GemFilter may stem from two factors: (1) the selection of important tokens, and (2) the re-computation of these tokens, which might mitigate issues like "lost-in-the-middle". To understand whether both factors made contributions, we provide an ablation study to isolate the contribution of each factor.

In Figure 10, we introduce GemFilter-One-Run, which does not have the second run as GemFilter. In detail, after getting the indices, which is exactly the same as GemFilter, it directly uses this index set to evict the KV cache for all attention heads and attention layers and continuously conducts the iterative generation phase.

### D.5.1 DIFFERENCE FROM GEMFILTER AND SNAPKV

It is different from GemFilter as (1) it requires computing full attention matrices for all layers for the KV cache eviction, so it does not save prompt computation phase complexity; (2) it does not have the second run so that the RoPE positional distance is not updated as GemFilter, where its distance between 'needle' and query can be very large.

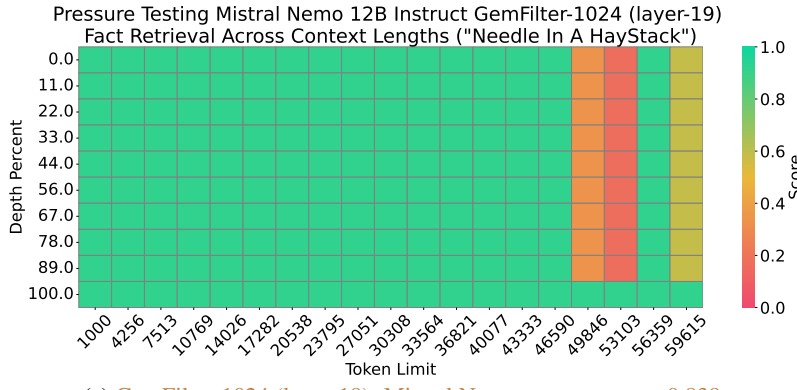

(a) GemFilter-1024 (layer-19). Mistral Nemo average score: 0.838.

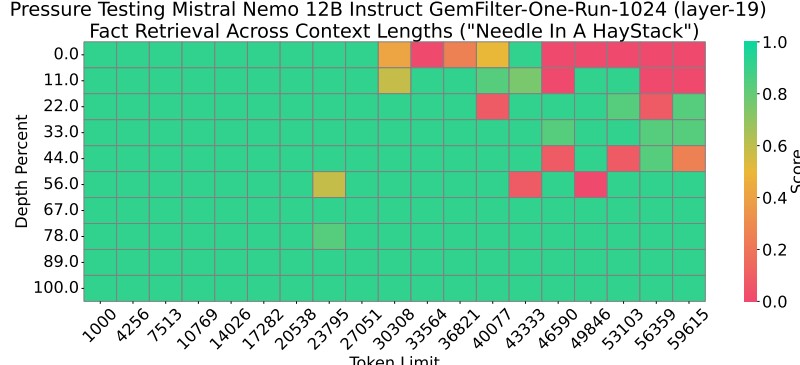

(b) GemFilter-One-Run-1024 (layer-19). Mistral Nemo average score: 0.827.

Figure 10: Needle in a Haystack performance comparison of different methods using the Mistral Nemo 12B Instruct model. The $x$-axis represents the length of the input tokens, while the $y$-axis shows the position depth percentage of the 'needle' information (e.g., 0% indicates the beginning, and 100% indicates the end). A higher score reflects better performance, meaning more effective retrieval of the 'needle' information. (a) is our method GemFilter and (b) is the degenerate version GemFilter-One-Run for ablation study.

It is different from SnapKV as all attention heads and attention layers share the same index set, while SnapKV has different index sets for different attention heads and different attention layers.

### D.5.2 RESULTS

As we can see in Figure 10, the GemFilter-One-Run has a comparable performance with GemFilter, while it is worse when the distance between the query and the 'needle' is large. This is expected as the RoPE positional distance does not update in GemFilter-One-Run. On the other hand, the GemFilter-One-Run takes a larger running time complexity and a larger memory consumption than GemFilter as it requires computing full attention matrices for all layers, while GemFilter only needs to compute the first few layers.

### D.6 INDEX SELECTION

In Figure 11, we visualize the top-$k$, $k = 100$, indices over length $n = 46,530$ of each attention layer in GemFilter and SnapKV when using the Mistral Nemo 12B Instruct model and evaluating on Needle in a Haystack. The GemFilter uses layer-19 as its filter layer. Recall that GemFilter selects the top-$k$ indices based on the summation of all attention heads, so each attention layer only has one index set. The SnapKV selects top-$k$ indices for each attention head, so each attention layer only has $h = 32$ index sets, where $h$ is the number of attention heads in each attention layer. Thus, for GemFilter and SnapKV, we plot 1 and 32 index sets for each attention layer, respectively.

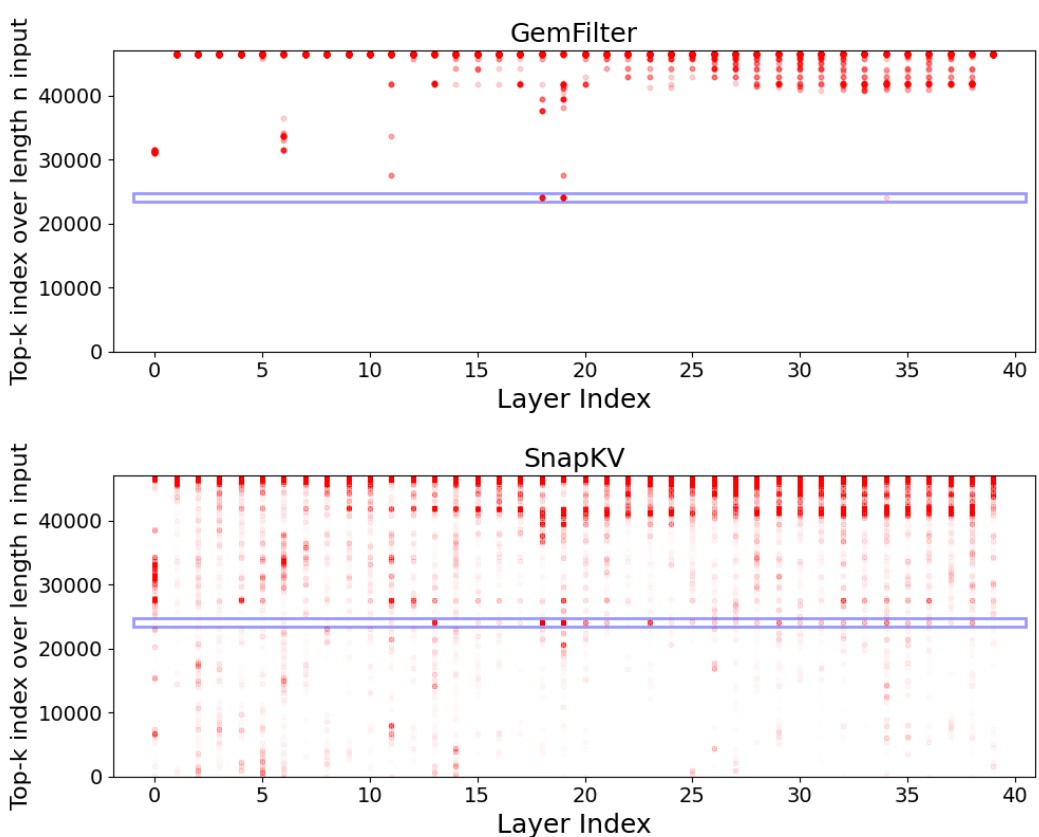

Figure 11: Needle in a Haystack visualization of the top-$k$ indices of each attention layer in GemFilter and SnapKV when using the **Mistral Nemo 12B Instruct** model. The GemFilter uses layer-19 (the same as other experiments) as its filter layer. Both GemFilter and SnapKV use $k = 100$, i.e., the number of selected tokens. The $x$-axis is the layer index, 40 layers in total. The $y$-axis is the input index, where the input token length is $n = 46,530$. We use 50% as the position depth percentage of the 'needle' information. The red dots mean the selected tokens for the corresponding layer and input tokens. The blue rectangle represents the position of the needle information. The output of GemFilter is *"The best thing to do in San Francisco is eat a sandwich and sit in Dolores Park on a sunny day."* which is totally correct. The output of SnapKV is *"The best thing to do in San Francisco is eat a sandwich."* which is partially correct.

In Figure 11, the red dots mean the selected tokens for the corresponding layer and input tokens. The blue rectangle represents the position of the needle information. The output of GemFilter is *"The best thing to do in San Francisco is eat a sandwich and sit in Dolores Park on a sunny day."* which is totally correct. The output of SnapKV is *"The best thing to do in San Francisco is eat a sandwich."* which is partially correct.

We can see that GemFilter is only focused on the needle information and recent information, while SnapKV focuses on a wide range of tokens, which may distract its attention. We can also conclude that GemFilter and SnapKV have very different selection mechanisms.

### D.7 LLaMA 3.1 Chat Template

In Table 3, we report the performance of different methods on the LongBench QA task using LLaMA 3.1 8B Instruct and its official LLaMA Chat template[11]. In the following, we show the PyTorch code of the way we use the LLaMA Chat template.

---

[11]https://huggingface.co/meta-llama/Llama-3.1-8B-Instruct

```
1 messages = [
2     {"role": "system", "content": ""},
3     {"role": "user", "content": prompt}]
4
5 input = tokenizer.apply_chat_template(messages, add_generation_prompt=
      True, return_tensors="pt", return_dict=True).to(device)
```

In Table 3, we can see that, after applying the template, all methods gain a large improvement in performance compared to Table 1. Also, we can see that GemFilter has a performance comparable to that of other state-of-the-art methods. It is interesting to understand the difference between the attention mechanisms with and without using a chat template. We leave it as our future work.

Table 3: Performance comparison on LongBench across various methods when using LLaMA 3.1 8B Instruct and its official LLaMA Chat template. A larger number means better performance. The best score is **boldfaced**.

| Method | Single-Document QA | | | Multi-Document QA | | | |
| | NrtvQA | Qasper | MF-en | HotpotQA | 2WikiMQA | Musique | Average |
|---|---|---|---|---|---|---|---|
| All KV | 25.08 | **44.06** | 55.08 | 47.86 | 49.19 | 27.46 | 41.46 |
| MInference | **29.61** | 43.89 | 54.76 | 51.72 | 49.55 | 28.17 | 42.95 |
| SnapKV-1024 | 29.01 | 41.67 | **56.22** | **56.81** | 49.32 | **31.56** | **44.10** |
| GemFilter-1024 | 22.8 | 40.78 | 48.05 | 54.33 | **50.03** | 30.03 | 41.00 |

## D.8 MORE RESULTS OF INDEX SELECTION

In this section, we provide more results of index selection on LLaMA 3.1 8B Instruct and Phi 3.5 Mini 3.8B Instruct, where the setting is similar as Figure 11.

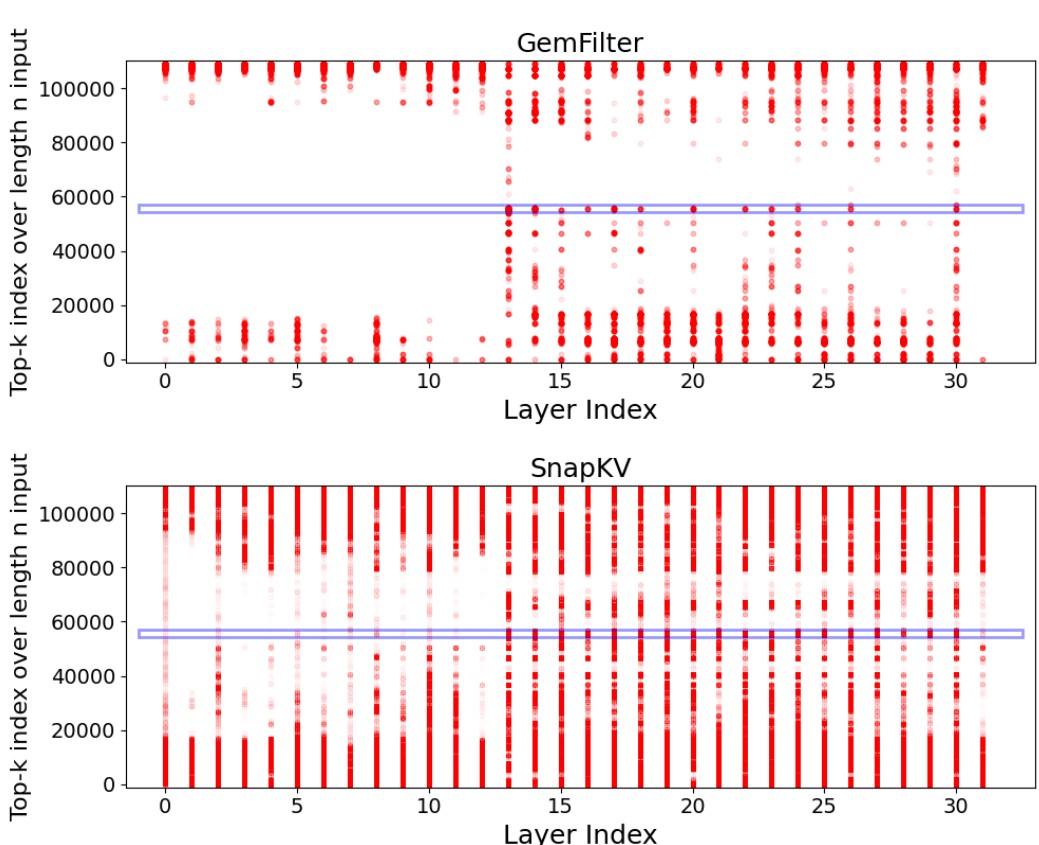

Figure 12: Needle in a Haystack visualization of the top-$k$ indices of each attention layer in GemFilter and SnapKV when using the **LLaMA 3.1 8B Instruct** model. The GemFilter uses layer-13 (the same as other experiments) as its filter layer. Both GemFilter and SnapKV use $k = 1024$, i.e., the number of selected tokens. The $x$-axis is the layer index, 32 layers in total. The $y$-axis is the input index, where the input token length is $n = 108,172$. We use 50% as the position depth percentage of the 'needle' information. The red dots mean the selected tokens for the corresponding layer and input tokens. The blue rectangle represents the position of the needle information. The output of GemFilter is *"Eat a sandwich and sit in Dolores Park on a sunny day."* which is totally correct. The output of SnapKV is *"Eat a sandwich at a deli in the Mission District."* which is partially correct.

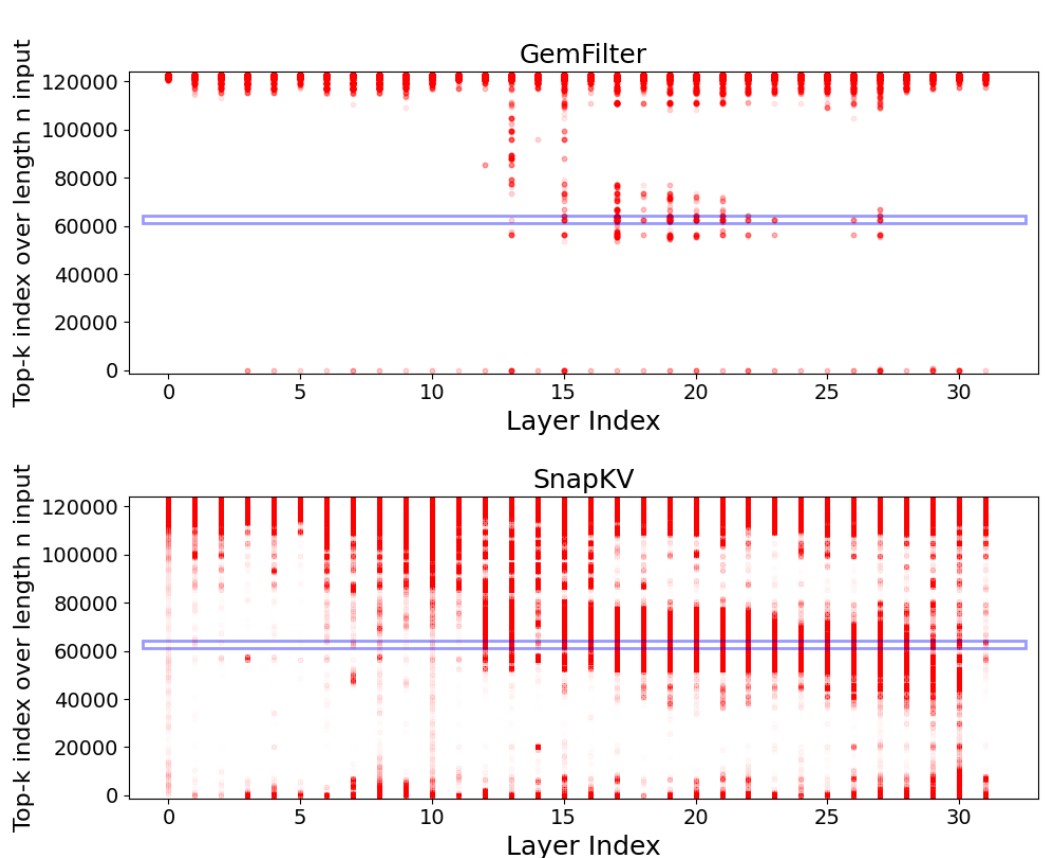

Figure 13: Needle in a Haystack visualization of the top-$k$ indices of each attention layer in Gem-Filter and SnapKV when using the **Phi 3.5 Mini 3.8B Instruct** model. The GemFilter uses layer-19 (the same as other experiments) as its filter layer. Both GemFilter and SnapKV use $k = 1024$, i.e., the number of selected tokens. The $x$-axis is the layer index, 32 layers in total. The $y$-axis is the input index, where the input token length is $n = 122,647$. We use 50% as the position depth percentage of the 'needle' information. The red dots mean the selected tokens for the corresponding layer and input tokens. The blue rectangle represents the position of the needle information. The output of GemFilter is *"Sit in Dolores Park on a sunny day and eat a sandwich."* which is totally correct. The output of SnapKV is *"Eat a sandwich."* which is partially correct.

