# OpenReview forum: "Discovering the Gems in Early Layers: Accelerating Long-Context LLMs with 1000x Input Token Reduction"
_ICLR.cc/2025/Conference — Submitted to ICLR 2025_

### Official Review · Reviewer_FAs6 · 2024-11-02

**Soundness:** 3
**Presentation:** 3
**Contribution:** 3
**Rating:** 6
**Confidence:** 3

**Summary:**

Stemming from the observation that only selecting 100 tokens in the pre-filling phase is enough to preserve the performance achieved by kv cache eviction methods, this paper proposes GemFilter that identifies relevant tokens with early layers and pass fewer tokens to the generation phase.

**Strengths:**

(1) While most KV Cache papers focus on the generation phase, this paper demonstrates the possibility of token eviction in the prefilling phase, which to my knowledge is interesting. Also, I am not actively working on KV cache eviction and might miss relevant references.

(2) The explanation of Algorithm 1 in section 3.2 is very clear.

(3) The flexible choice of filter layer is quite useful in practice, avoiding extensive hyperparameter tuning. I suspect this is highly related to the pre-LN used in most LLMs. I am wondering if it is possible to report a similar study of the filter layer for Gemma 2, as it uses a different layer norm.

**Weaknesses:**

(1) What is the intuition behind selecting tokens with the most attention specifically from the last query? I guess this will likely focus on the initial tokens and the last tokens that are close to the query. I would like to see more ablation studies here. Instead of the last query, can we choose other queries?  Baselines here can be randomly selecting rows of the attention matrix, and selecting rows with the largest l2 norm.

(2) A certain level of performance loss is observed for Phi 3.5 Mini. While for larger models like LLaMA 8B and Mixtral 12B, we somehow observe improved average scores. Perhaps, full tokens are still needed for small models.

(3) I am not sure how important to reduce the memory requirement for the pre-filling phase, as this phase is more computing-intensive than memory-intensive. And the computation here can be done in parallel. In contrast, memory is a big issue in the decoding phase.

(4) A very important baseline is missing: MInference (https://github.com/microsoft/MInference), which also aims to reduce pre-filling memory.

**Questions:**

1. The flexible choice of filter layer is quite useful in practice, avoiding extensive hyperparameter tuning. I suspect this is highly related to the pre-LN used in most LLMs. I am wondering if it is possible to report a similar study of the filter layer for Gemma 2, as it uses a different layer norm.

2. What is the intuition behind selecting tokens with the most attention specifically from the last query? I guess this will likely focus on the initial tokens and the last tokens that are close to the query. I would like to see more ablation studies here. Instead of the last query, can we choose other queries?  Baselines here can be randomly selecting rows of the attention matrix, and selecting rows with the largest l2 norm.

3. A certain level of performance loss is observed for Phi 3.5 Mini. While for larger models like LLaMA 8B and Mixtral 12B, we somehow observe improved average scores. Perhaps, full tokens are still needed for small models.

4. I am not sure how important to reduce the memory requirement for the pre-filling phase, as this phase is more computing-intensive than memory-intensive. And the computation here can be done in parallel. In contrast, memory is a big issue for the decoding phase.

---

> ### Author Response · Authors · 2024-11-22
>
> We extend our gratitude to the reviewer for their meticulous feedback. We offer the following elucidations:
>
>
> ### W1 & Q2: I would like to see more ablation studies here. Instead of the last query, can we choose other queries?
> Thank you for your valuable suggestions! We update the experiments with the two baselines you suggested in Section 4.4 and Appendix D.4 in revision Line 476-482 & 860-901.
>
> In Figure 9, we introduce two methods:  (a) selecting middle rows of the attention matrix and (2) selecting rows with the largest $\ell_2$ norm. Both methods fail in the Needle in a Haystack task, verifying that selecting the last query token is essential.
>
> ### W2 & Q3: A certain level of performance loss is observed for Phi 3.5 Mini. While for larger models like LLaMA 8B and Mixtral 12B, we somehow observe improved average scores. Perhaps, full tokens are still needed for small models.
> Thank you for your careful check! We can see that, in Table 1, with more tokens preserved, the GemFilter will have a better performance on Phi3.5, i.e., GemFilter-4096 is much better than GemFilter-1024. Note that the small model is fast and consumes light memory. Then, how to balance the trade-off between token number and performance is an interesting and important topic. We leave it as our future work.
>
> ### W3 & Q4: I am not sure how important to reduce the memory requirement for the pre-filling phase, as this phase is more computing-intensive than memory-intensive. And the computation here can be done in parallel. In contrast, memory is a big issue in the decoding phase.
>
> Thank you for your comments! We agree with your insightful opinion.
>
> Our method can save both memory consumption and running time in both pre-filling phase and decoding phase, i.e., we save all four complexities, as shown in Figure 3 and Figure 6. Thus, we believe that this is a strength of our method rather than a weakness.
>
> ### W4: A very important baseline is missing: MInference (https://github.com/microsoft/MInference), which also aims to reduce pre-filling memory.
> Thank you so much for pointing out these suggestions! We miss this important baseline in the original version. We have added this brilliant work in our revision Line 157-160. We also provide the comparison with Minference on LongBench [2] in Table 1, where our method is compatible with Minference, but faster than Minference in the prompt computation phase. We refer the reviewer to the **Missing baseline** section of Global Response for more details.
>
> We only evaluate MInference on LLaMA 3.1 8B Instruct as it is not supported on the Mistral Nemo 12B Instruct and Phi 3.5 Mini 3.8B Instruct currently.
>
> ### Q1: I am wondering if it is possible to report a similar study of the filter layer for Gemma 2, as it uses a different layer norm.
> Thank you for your question! We checked the official Gemma 2, which only has a context window of 8k, while our paper focuses on the long context settings. We will evaluate Gemma 2 in our next revision if the official Gemma 2 model supports a long context.
>
> We hope our new experiments address your concerns.
>
> ### Reference
>
> [1] Jiang, H., Li, Y., Zhang, C., Wu, Q., Luo, X., Ahn, S., ... & Qiu, L. (2024). Minference 1.0: Accelerating pre-filling for long-context llms via dynamic sparse attention. NeurIPS’24.

---

> > ### Comment · Reviewer_FAs6 · 2024-11-26
> > **Thank you for the response**
> >
> > I thank the authors for the experiments on larger models and the MInference baseline. Can the authors further explain why GemFilter falls short of MInference specifically in Single-Document QA?

---

> > > ### Author Response · Authors · 2024-11-27
> > > **Thank you and reply to new question**
> > >
> > > Thank you so much for your insightful question!
> > >
> > > In Table 1, we can see that GemFilter has a good performance on both Single-Document QA and Multi-Document QA when using Mistral Nemo or Phi 3.5, while GemFilter has drops in Single-Document QA compared to Minference.
> > >
> > > Note that Minference uses three types of attention sparsity, i.e., $\land$-shape, vertical-slash, and block-sparse attention head. Our conjecture is that, for Single Document QA, block-sparse attention head may have a better performance, particularly for LLaMA, as the “related information’’ may concentrate in continuous paragraphs. We will study the different attention pattern mechanisms in our follow-up works. Thanks for your valuable comments.
> > >
> > > [1] Jiang, H., Li, Y., Zhang, C., Wu, Q., Luo, X., Ahn, S., ... & Qiu, L. (2024). Minference 1.0: Accelerating pre-filling for long-context llms via dynamic sparse attention. NeurIPS’24.

---

> > > > ### Comment · Reviewer_FAs6 · 2024-12-01
> > > > **Thanks for the response.**
> > > >
> > > > I sincerely thank the authors for their reply. I am happy to maintain my original division of week acceptance.

---

> > > > > ### Author Response · Authors · 2024-12-01
> > > > > **Thank you**
> > > > >
> > > > > We thank the reviewer's valuable comments and positive score. We appreciate your time and response!

---

### Official Review · Reviewer_Np9u · 2024-11-02

**Soundness:** 2
**Presentation:** 2
**Contribution:** 2
**Rating:** 5
**Confidence:** 4

**Summary:**

This paper introduces GemFilter for accelerating large language models (LLMs) when processing long-context inputs. The key insight is that LLMs can identify important tokens in their early layers before generating answers, which allows for significant input compression. The proposed GemFilter algorithm uses early layers of an LLM as filters to select and compress input tokens, reducing the context length for subsequent processing. The method achieves a 2.4× speedup and 30% reduction in GPU memory usage compared to state-of-the-art methods like SnapKV/H2O. Results show that  GemFilter outperforms standard attention and SnapKV in the Needle in a Haystack test, while maintaining comparable performance on the LongBench benchmark.

**Strengths:**

1) This paper focuses on a timely and important problem: Accelerating Long-context LLM inference.

2) The method introduced in GemFilter is simple-yet-effective, according to the experimental results provided in the paper (better accuracy than the dense baseline on the NIAH task, comparable performance on LongBench evaluation).

3) GemFilter achieves a 2.4× speedup and 30% reduction in GPU memory usage compared to state-of-the-art methods like SnapKV/H2O.

**Weaknesses:**

1) GemFilter only uses the attention score of the last token during the context/prefilling stage to determine the important tokens to keep. And then uses the selected tokens to run the full inference process. The process may lose some information in the context. For instance, for multi-hop Q&A tasks/multi-needle in a haystack evaluation, there will be questions focusing on different parts of the given long context. I am wondering if GemFilter will still be able to keep the model's performance on such tasks.

2) The selection of the filter layer seems to be results-oriented. And there seems to be no specific method to efficiently identify the filter layer. In Table 2, it seems that there is prominent performance loss on some benchmarks when choosing layer-13 as the filter layer (e.g., NrtvQA, Qasper, MF-en). The robustness of the method still needs to be further demonstrated.

3) For Mistral Nemor and Phi 3.5, the filter layer appears in the middle (or even later) of the model, limiting the efficiency gain that could be achieved with this method.

**Questions:**

1) The LongBench evaluation scores for the dense attention baseline (in Table 1) seems to be lower than expectation. For example, the llama-3-8B model only achieves a score of 13.04 on qasper with full KV.

2) In Figure 4, GemFilter achieves a much higher score than the dense baseline on the NIAH test. I would appreciate it if the authors could provide some analysis for the improvements above the dense attention.

3) How is the running time of decoding stage (gen time) computed in Figure 3/6? What is the specific setting for the evaluation (i.e., generation length)?

---

> ### Author Response · Authors · 2024-11-22
>
> We extend our gratitude to the reviewer for their meticulous feedback. We offer the following elucidations:
>
> ### W1: multi-hop Q&A task
> Thank you for your question. Note that the LongBench benchmark has multi-hop QA tasks like HotpotQA and 2WikiMQA. Our method beats all other baselines in Table 1, which shows that GemFilter has a good ability to handle questions focusing on different parts of the given long context. We find that GemFilter is able to select key information from multiple documents rather than only focusing on one part.
>
> ### W2: The robustness of the method still needs to be further demonstrated.
> Thank you for your careful check! We agree that for different tasks the best filter layer may be different. However, we show that some layers have competitive overall tasks, although not the best. On the other hand, we can adaptively choose the layer $r$ on a given task:
> - For a fixed $r$, we use the $r$-th layer as the filter to generate the output. Compare the output with filtering and the output without filtering, and compute the similarity as a metric.
> - Then, we pick the $r$ that leads to the largest similarity.
>
> We also point out that one can use traditional techniques for hyperparameter selection for picking $r$, such as using a validation set from the target task. Note that hyperparameters are typical in machine learning methods. Our method has only one hyperparameter, and there is a simple rule of thumb for setting it and a good value range to get stable performance, which is also believed as a strength point from reviewer FAs6. Thus, we believe setting the hyperparameter $r$ is not difficult.
>
> ### W3: For Mistral Nemor and Phi 3.5, the filter layer appears in the middle (or even later) of the model, limiting the efficiency gain that could be achieved with this method.
> Thank you for your comments. Our method reduces both the prompt computation and iterative generation time.
> - Filtering using the middle layer or nearby still reduces the prompt time by nearly half (see Figure 6).
> - Furthermore, note that the reduction in the generation time is not affected by the filter layer selection. The reduction is significant, i.e., 2x to 5x speedup.
>
> On the other hand, as our method reduces both the prompt computation and iterative generation time, reviewer FAs6 values our novelty that “While most KV Cache papers focus on the generation phase, this paper demonstrates the possibility of token eviction in the prefilling phase, which to my knowledge is interesting.”
>
> ### Q1: Baseline (in Table 1) seems to be lower than expectation
> Thank you for your careful reading. We also noted that Llama-3-8B does not perform well on some QA tasks. The reason is that we have not used the chat template specialized for Llama. As we have not used the chat template for Mistral Nemo and Phi 3.5, for a fair comparison, we just keep the same protocol among all models.
>
> ### Q2: Some analysis for the improvements above the dense attention
> Thank you for your valuable questions! There are two reasons that our method is better than dense attention:
> - Our method's filtering already helps filter out most useless tokens. This significantly helps the LLMs that were otherwise lost in the haystack.
> - In the second run of GemFilter, the input length will be reduced from $n$ to $k$. The RoPE positional distance among the tokens will be much smaller, and the LLMs have better performance in a short context.
>
> On the other hand, we add new ablation studies in Section 4.4 and Appendix D.5 in the revision (Line 483-485 & 903-962) to further verify our second points above. We refer the reviewer to our revision for more details.
>
> ### Q3: How is the running time of decoding stage (gen time) computed in Figure 3/6? What is the specific setting for the evaluation (i.e., generation length)?
>
> Thank you for your question! The iterative generation running time and memory consumption are evaluated using $50$ tokens generation. We add the setting for the evaluation of running time and memory consumption in the revision Line 121-122 & 489-490.
>
> We hope our response addresses your concerns.

---

> > ### Comment · Reviewer_Np9u · 2024-11-23
> >
> > I would like to thank the authors for the detailed response. And I still have a few follow-up concerns regarding the clarifications provided:
> >
> > Regarding W1: I would like to see more clarification on the token filtering process in multi-hop question-answering scenarios, particularly in interactive settings. When questions are presented sequentially, particularly in a multi-round dialogue format rather than simultaneously, does the token filtering process need to be repeated for each new query? This raises a potential concern about information loss, as tokens that might be crucial for answering future questions could be inadvertently pruned during filtering steps in earlier rounds of conversation.
> >
> > Regarding W2: While the authors propose adaptively choosing layer $r$ for specific tasks to address accuracy degradation, I believe the robustness concerns persist for several reasons. First, in the context of deploying a general-purpose LLM, it may be impractical to calibrate hyperparameters in advance given the diverse nature of user requests. Second, the hyperparameter search process itself incurs computational overhead. The task-specific nature of these hyperparameters could potentially limit the method's practical applicability in real-world scenarios where task requirements are not known a priori.
> >
> > Regarding Q1: I have concerns about the evaluation methodology when the chat template differs from the model's original template. In cases where the model's baseline performance is already much lower than expectation, the evaluation may be less informative / convincing. The accuracy loss caused by the proposed method may not be accurately reflected, since the accuracy of the original model is already impaired due to the template mismatch.

---

> > > ### Author Response · Authors · 2024-11-25
> > > **Thank you and further reply to new concerns (Part 1)**
> > >
> > > We are glad that our reply has addressed some concerns from the reviewer! We sincerely thank you for your insightful response. We appreciate your time and we would like to fix your follow-up concerns.
> > >
> > > ### W1.1: In a multi-round dialogue format rather than simultaneously, does the token filtering process need to be repeated for each new query?
> > > Thank you so much for your insightful questions! Here is the way to use GemFilter for multi-round dialogue.
> > >
> > > We use LLaMA 3.1 8B Instruct as an example. After each round of dialogue, we keep the full KV Cache of the whole history for the first 13 layers. For a new coming query, we use the first 13 layers to get the index set and then conduct the second run of GemFilter. Finally, we update the first 13 layers’ KV Cache for the new round.
> > >
> > > Thus, when $n \gg k$, compared to the standard way:
> > > - The KV Cache memory consumption of GemFilter is only 13/32 of the KV Cache memory consumption of the standard way.
> > > - The running time is only 13/32 of the running time of the standard way, as we only compute the new query over the full KV Cache over 13 layers rather than 32 layers.
> > >
> > > ### W1.2: This raises a potential concern about information loss, as tokens that might be crucial for answering future questions could be inadvertently pruned during filtering steps in earlier rounds of conversation.
> > > Thank you for your pointing out this! We agree that this is a certain concern in the KV-cache compression community. Indeed, most static KV-cache compression methods in this line, e.g., H2O [1], SnapKV [2], and MInference [3], suffer from this information loss problem.
> > >
> > > Thus, some very recent preprint work [4,5], which was released a few days before the ICLR 2025 submission deadline, tried to solve the problem by using dynamic KV-cache compression. They need to save all layers full KV-cache in the memory and use the approximate nearest neighbor search method, e.g,  IVF indexes in [4] and Local Sensitivity Hashing in [5], for dynamic queries. However, there are several concerns.
> > > - There is no efficient GPU implementation of the approximate nearest neighbor search method. Thus, they need to move part of the computation to the CPU, which may introduce additional IO and communication consumption.
> > > - They are not memory-efficient as they need to save all layers' full KV-cache. GemFilter only needs to store early layers’ full KV-cache.
> > > - Dynamic KV-cache compression is a very new direction and the effectiveness of many methods has not been fully verified by the community.
> > >
> > > Extending GemFilter to a dynamic KV-cache compression setting is very interesting. It may be beyond the scope of this paper and the workload of its implementation definitely is over rebuttal. We are leaving it on our TODO list. We are willing to discuss more per the reviewer's request. Thank you again for your constructive feedback!
> > >
> > > ### W2:  The task-specific nature of these hyperparameters could potentially limit the method's practical applicability in real-world scenarios where task requirements are not known a priori.
> > > Thank you so much for your questions. We try to address your question below:
> > > - If we know downstream tasks, e.g., a chatbot for a specific use, we can turn this hyperparameter according to our previous response.
> > > - If we do not know the downstream task, we can choose some layer that is generally good, e.g., 13 layer for LLaMA 3.1 8B Instruct and 19 layer for Mistral Nemo 12B Instruct. Overall, their performance is not bad in all tasks as shown in LongBench.
> > >
> > > On the other hand, [1,2,3] also have hyper-parameters and may meet the same issues. We also would like to point out that knowledge-free acceleration or adaptation may be a hard topic, also known as the no-free lunch Theorem. For example, instruction finetuning may make the model loss some ability.
> > >
> > > We hope our proposed solution and discussion may relieve your concerns.
> > >
> > > ### Q1: The accuracy loss caused by the proposed method may not be accurately reflected, since the accuracy of the original model is already impaired due to the template mismatch.
> > > Thank you so much for your suggestion.
> > >
> > > We report the performance of different methods on the LongBench QA task using LLaMA 3.1 8B Instruct and its official LLaMA Chat template. We refer reviewer to Line 1020-1047 of the second version revision for more details.
> > >
> > > In Table 3 of the revision, we can see that, after applying the template, all methods gain a large improvement in performance compared to Table 1. Also, we can see that GemFilter has a performance comparable to that of other state-of-the-art methods.
> > > It is interesting to understand the difference between the attention mechanisms with and without using a chat template. We leave it as our future work.
> > >
> > > We hope our answer may address your new concerns.

---

> > > ### Author Response · Authors · 2024-11-25
> > > **Thank you and further reply to new concerns (Part 2)**
> > >
> > > ### Reference
> > >
> > > [1] Zhang, Z., Sheng, Y., Zhou, T., Chen, T., Zheng, L., Cai, R., ... & Chen, B. (2023). H2o: Heavy-hitter oracle for efficient generative inference of large language models. NeurIPS’23.
> > >
> > > [2] Li, Y., Huang, Y., Yang, B., Venkitesh, B., Locatelli, A., Ye, H., ... & Chen, D. (2024). Snapkv: Llm knows what you are looking for before generation. NeurIPS’24.
> > >
> > > [3] Jiang, H., Li, Y., Zhang, C., Wu, Q., Luo, X., Ahn, S., ... & Qiu, L. (2024). Minference 1.0: Accelerating pre-filling for long-context llms via dynamic sparse attention. NeurIPS’24.
> > >
> > > [4] Liu, D., Chen, M., Lu, B., Jiang, H., Han, Z., Zhang, Q., ... & Qiu, L. (2024). Retrievalattention: Accelerating long-context llm inference via vector retrieval. arXiv preprint arXiv:2409.10516.
> > >
> > > [5] Chen, Z., Sadhukhan, R., Ye, Z., Zhou, Y., Zhang, J., Nolte, N., ... & Chen, B. (2024). Magicpig: Lsh sampling for efficient llm generation. arXiv preprint arXiv:2410.16179.

---

> ### Comment · Reviewer_Np9u · 2024-11-29
>
> Thank you for the response!
>
> For W1, I agree with the authors that one can run GemFilter for multiple times when dealing with multi-round dialogues. Therefore, if the questions are provided in different conversation rounds, GemFilter may still be able to locate the related contexts. However, assume that GemFilter will use the first 13 of 32 layers to filter the important tokens, when there is more than 3 rounds of dialogue, the total prefilling time would be (n * 13/32) times slower than the dense baseline, where n is the number of conversation rounds.
>
> For Q1, I would like to thank the authors for providing new evaluation results with official LLaMA Chat template. I think the results in Line 1020-1047 can more accurately capture the GemFilter's actual performance now (compared to Table 1).

---

> > ### Author Response · Authors · 2024-11-29
> > **Thank you, and more clarification for W1**
> >
> > We are glad that our response fixes some of the reviewer's concerns. We sincerely thank you for your time and effort. We would like to clarify the new concerns.
> >
> > Note that when the context is long, the running time complexity bottleneck is full attention matrix computation and memory of full KV cache size.
> >
> > For multi-round dialogues, where the round number is $r$ and the context length is $n$:
> > - The dense baseline needs to run 32-layer full attention matrix computation and save 32-layer full KV cache size, which is **independent with $r$**.
> > - The GemFilter needs to run 13-layer full attention matrix computation and save 13-layer full KV cache size, which is **independent with $r$** as well. Note that we do not need to re-prefilling the first 13 layers as we already saved the results in KV cache, the same as the dense baseline. Also, note that the last row of the attention matrix can be directly computed from the KV cache and new query.
> >
> > Below, we give a detailed example. Assume length $n$ context $T$ is super long, and the $Q_1$ is the query. We can use GemFilter to get answer $A_1$, where the first 13 layer-th attention of KV($[T,Q_1]$) is saved in the GPU memory. Then, we compute the KV cache of KV([$T,Q_1, A_1$]), which is efficient based on KV($[T,Q_1]$) and only takes $O(n)$ time as the dense baseline.
> > - Note that desne baseline need to take $O(n)$ during generation decoding, while GemFilter takes $O(n)$ after $A_1$ generated. Thus, this $O(n)$ time can be run off-line during the user reads $A_1$ and types $Q_2$. The generation decoding complexity of GemFilter is $O(k)$, e.g., $k=1024$.
> > Then, after user type $Q_2$, we can use the same strategy get  KV([$T, Q_1, A_1, Q_2$]) based on the KV([$T,Q_1, A_1$]) efficiently, e.g., $O(n)$. Then, the GemFilter can generate the answer $A_2$.
> >
> > Note that during the whole pipeline above, the full attention matrix computation of the first 13 layers only has once. Thus, compared with the dense baseline, the time and memory complexity of GemFilter is only 13/32 for prefilling and $k/n$ for generation decoding, which is independent of the round number $r$.
> >
> > We hope that our clarification can relieve your concerns. We are willing to discuss more if the reviewer has more follow-up questions.

---

> > > ### Comment · Reviewer_Np9u · 2024-11-29
> > >
> > > Thank you for the clarification! Now I understand that the total prefilling time of GemFilter would be still shorter than the dense baseline when dealing with multi-round dialogues. I raised my score to 5.

---

> > > > ### Author Response · Authors · 2024-11-30
> > > > **Thank you**
> > > >
> > > > We are glad that our response fixes your concerns! If the reviewer has any further concerns, we are willing to address them. We appreciate the reviewer for the time and effort involved in the review.

---

### Official Review · Reviewer_rvyU · 2024-11-03

**Soundness:** 2
**Presentation:** 3
**Contribution:** 2
**Rating:** 5
**Confidence:** 4

**Summary:**

This paper looks into KV cache compression for efficient LLM inference under long context. The key insight is that LLMs can identify important tokens in early layers before generating answers. As such, the paper proposes to use early layers to select important tokens, effectively reducing the context length for subsequent generation. Evaluation shows that the proposed method outperforms SnapKV and H2O with additional memory reduction and speedups.

**Strengths:**

- Interesting observation that the last row of attention matrices in early layers may serve as a signal for locating important tokens.
- Promising results with good accuracy and memory savings.

**Weaknesses:**

- Limited technical novelty. Prior work has identified that top layers of LLMs are not very effective, https://arxiv.org/abs/2403.17887, e.g, by pruning up to half of the layers, the model remain accurate. This seems to be more of an issue of the public LLMs that are not well-trained to leverage those deeper layers. From that perspective, the proposed method is more of a hack that tackles these issues, which may disappear as the pre-training recipe evolves.

- It may not be easy to robustly choose the layer r for different tasks and models. While using early layers to identify a fixed set of tokens, the method may lose adaptiveness in KV cache compression. As a result, it may suffer from poor generalizability on different tasks, e.g., those that are not tested in the paper. Therefore, for this particular work, more datasets should be included for evaluation.

**Questions:**

In practice, LLMs are generalists, and therefore need to handle different tasks and requests dynamically. How do you make sure that the proposed method can robustly find a good early-layer signal to get high accuracy for a mixture of tasks and requests.

---

> ### Author Response · Authors · 2024-11-22
>
> We extend our gratitude to the reviewer for their meticulous feedback. We offer the following elucidations:
>
> ### W1: Public LLMs that are not well-trained to leverage those deeper layers
> Thank you for pointing out these insightful comments. The author has a different opinion from the reviewer. We think the "not well trained" argument might not be justified/true with current LLMs, particularly note that LLaMA3.1 has been trained on 15T tokens.
>
> From our perspective, the phenomenon that early layers can be used as filters is not because the LLMs are not well-trained. They are well-trained, such that the early layers are more responsible for understanding and analyzing the input while the later layers are more responsible for generating outputs. From this perspective, such property is inherent in well-pretrained language models. While things can change when completely new paradigms are proposed, we believe our techniques are significant contributions to the community.
>
> ### W2.1: It may not be easy to robustly choose the layer $r$ for different tasks and models.
> Thank you for pointing this out! Due to the limited time of the rebuttal period, we are not able to fully solve this problem by some end-to-end automatic methods. We have some possible adaptive methods to choose the layer $r$ on a given task:
> - For a fixed $r$, we use the $r$-th layer as the filter to generate the output. Compare the output with filtering and the output without filtering, and compute the similarity as a metric.
> - Then, we pick the $r$ that leads to the largest similarity.
>
> On the other hand, we can simply pick some layer near the middle layer, which then leads to good performance as shown in Table 2. Thus, the rule of thumb is to set $r$ to be close to half of the number of layers.
>
> We also point out that one can use traditional techniques for hyperparameter selection for picking $r$, such as using a validation set from the target task. Note that hyperparameters are typical in machine learning methods. Our method has only one hyperparameter, and there is a simple rule of thumb for setting it and a good value range to get stable performance, which is also believed as a strength point from reviewer FAs6. Thus, we believe setting the hyperparameter $r$ is not difficult.
>
> ### W2.2: More datasets should be included for evaluation.
> We believe that LongBench may be already a good representative set of tasks or datasets, where it is a multi-task benchmark designed to rigorously evaluate long-context understanding capabilities across various datasets, including single- and multi-document Question Answering (QA), summarization, few-shot learning, and synthetic tasks.
>
> Due to the limited time of the rebuttal period, we are not able to fully examine our methods on a new benchmark. However, we have run some new ablation studies (Line 476-485 & 860-962) and compared our method with more baselines (Line 378-407 & 419-424 & 430-431). We refer the reviewer to our revision for more details.
>
> ### Q1: In practice, LLMs are generalists, and therefore need to handle different tasks and requests dynamically. How do you make sure that the proposed method can robustly find a good early-layer signal to get high accuracy for a mixture of tasks and requests.
>
> Thank you for your questions! On the one hand, in our experiments, we use a fixed filter layer for all experiments, and we can see the results are great overall tasks. Thus, we believe that our filter layer is robust. On the other hand, our conjecture is that the LLMs use early layers to understand and analyze the input while the later layers are more responsible for generating outputs. To fully verify the conjecture requires substantial efforts and may be beyond the scope of this work. We leave this interesting and important topic as our future direction.
>
>
> We hope our response addresses your concerns.

---

> > ### Author Response · Authors · 2024-12-02
> > **Looking forward to receiving your feedback**
> >
> > Dear Reviewer rvyU,
> >
> > We hope we have adequately addressed your issues. We would be very grateful if you could provide feedback on our rebuttal since the discussion deadline is approaching in one day. If you require further clarification or have any additional concerns, please do not hesitate to contact us. We are more than willing to continue communicating with you.
> >
> > Warmest regards,
> >
> > Authors

---

> > > ### Comment · Reviewer_rvyU · 2024-12-02
> > > **Post-rebuttal response**
> > >
> > > Dear authors,
> > >
> > > Thank you for providing the response! The work has a good potential but the current state of the draft leaves multiple questions related to the robustness and generalizability of the proposed method on the table, such as the selection of r across different tasks and the benefit it brings to different models. Also, regardless of whether LLMs are well-trained or not, it would be better if the authors include simple baselines such as pruning of top layers like in https://arxiv.org/abs/2403.17887 with varying pruning ratio. This may make readers more appreciate the contribution of the work.

---

> > > > ### Author Response · Authors · 2024-12-04
> > > > **Thank you**
> > > >
> > > > Thank you for your reply. We appreciate your valuable time.
> > > >
> > > > For the robustness problem, the authors do not think it is an issue, as our response stated in **W2.1**. For the new experiments about the pruning ratio, as the time is very close to the end of the rebuttal, we could not finish that. We sincerely thank you for the valuable suggestions, and we will try to add them to the next version.

---

### Official Review · Reviewer_bddG · 2024-11-04

**Soundness:** 3
**Presentation:** 3
**Contribution:** 3
**Rating:** 5
**Confidence:** 4

**Summary:**

This paper introduces GemFilter, which leverages early layers of large language models (LLMs) to compress input tokens and accelerate inference. GemFilter identifies relevant tokens in the early layers before generating answers to a query, significantly reducing the context length for subsequent processing. The proposed method achieves a 2.4 $\times$ speedup and 30% reduction in GPU memory usage compared to SOTA methods.

**Strengths:**

This paper is targeting a critical issue in large language models (LLMs) by proposing a novel approach to accelerate inference and reduce GPU memory consumption. The proposed method, GemFilter, demonstrates substantial improvements in both speed and memory efficiency compared to existing techniques, such as standard attention and SnapKV/H2O. The paper is well-written and easy to follow, and the experimental results are convincing. The evaluation on the Needle in a Haystack task shows that GemFilter significantly outperforms standard attention, SnapKV and demonstrates comparable performance on the LongBench challenge. The proposed method is simple, training-free, and broadly applicable across different LLMs. The paper also provides interpretability by allowing humans to inspect the selected input sequence.

**Weaknesses:**

The method presented in the paper lies between prompt compression and KV cache compression. Its token selection approach aligns more closely with KV cache compression techniques like SnapKV and H2O, while its re-computation of selected tokens resembles prompt compression methods such as LLMLingua [1] and LongLLMLingua [2]. Although the authors compare their method with SnapKV and H2O, including a comparison with prompt compression methods, particularly LongLLMLingua, would enhance the analysis.

The performance improvement of GemFilter may stem from two factors: (1) the selection of important tokens, and (2) the re-computation of these tokens, which might mitigate issues like "lost-in-the-middle." An ablation study to isolate the contribution of each factor would be beneficial.

[1] Jiang, Huiqiang, et al. "Llmlingua: Compressing prompts for accelerated inference of large language models." arXiv preprint arXiv:2310.05736 (2023).

[2] Jiang, Huiqiang, et al. "Longllmlingua: Accelerating and enhancing llms in long context scenarios via prompt compression." arXiv preprint arXiv:2310.06839 (2023).

**Questions:**

Please refer to the weaknesses section for questions. I am willing to adjust the score if the concerns are addressed.

---

> ### Author Response · Authors · 2024-11-22
>
> We extend our gratitude to the reviewer for their meticulous feedback. We offer the following elucidations:
>
> ### W1: Including a comparison with prompt compression methods, particularly LongLLMLingua, would enhance the analysis.
> Thank you so much for pointing out these suggestions! We miss this important line of work [1,2,3] in the original version. We have added these brilliant works in our revision Line 174-179. We also provide a comparison with Llmlingua on LongBench [2] in Table 1, where our method outperforms Llmlingua [1]. We refer the reviewer to the **Missing baseline** section of Global Response for more details.
>
> We skip LongLLMLingua for a fair comparison, as LongLLMLingua requires explicitly separating the input context into text information and questions, while other methods do not require that.
>
>
> ### W2: An ablation study to isolate the contribution of each factor would be beneficial.
> Thank you so much for your valuable suggestions! We have added the corresponding ablation studies in Section 4.4 and Appendix D.5 in revision Line 483-485 & 903-962.
>
> In Figure 10, we introduce GemFilter-One-Run, which does not have the second run as GemFilter. In detail, after getting the indices as GemFilter, it directly uses this index set to evict the KV cache for all attention heads and attention layers and continuously conducts the iterative generation phase.
>
> **Difference from GemFilter and SnapKV.** It is different from GemFilter as (1) it requires computing full attention matrices for all layers for the KV cache eviction, so it does not save prompt computation phase complexity; (2) it does not have the second run so that the RoPE positional distance is not updated as GemFilter, where its distance between `needle' and query can be very large.
>
> It is different from SnapKV as all attention heads and attention layers share the same index set, while SnapKV has different index sets for different attention heads and different attention layers.
>
> **Results.** As we can see in Figure 10, the GemFilter-One-Run has a comparable performance with GemFilter, while it is worse when the distance between the query and the `needle' is large. This is expected as the RoPE positional distance does not update in GemFilter-One-Run. On the other hand, the GemFilter-One-Run takes a larger running time complexity and a larger memory consumption than GemFilter as it requires computing full attention matrices for all layers, while GemFilter only needs to compute the first few layers.
>
> We hope our new experiments address your concerns.
>
> ### Reference
>
> [1] Jiang, H., Wu, Q., Lin, C. Y., Yang, Y., & Qiu, L. (2023). Llmlingua: Compressing prompts for accelerated inference of large language models. EMNLP’23.
>
> [2] Jiang, H., Wu, Q., Luo, X., Li, D., Lin, C. Y., Yang, Y., & Qiu, L. (2023). Longllmlingua: Accelerating and enhancing llms in long context scenarios via prompt compression. ACL’24.
>
> [3] Pan, Z., Wu, Q., Jiang, H., Xia, M., Luo, X., Zhang, J., ... & Zhang, D. (2024). Llmlingua-2: Data distillation for efficient and faithful task-agnostic prompt compression. ACL’24.

---

> > ### Comment · Reviewer_bddG · 2024-11-25
> >
> > I extend my gratitude to the authors for their comprehensive response, particularly the inclusion of additional baselines and ablation studies of *GemFilter-One-Run*. I have a follow-up question:
> >
> > - In the GemFilter methodology, the initial $r$ layers are employed to identify significant tokens, whereas in GemFilter-One-Run, the entirety of the $m$ layers is utilized. Could you provide insights into the consistency of this approach? Specifically, are the tokens selected by the initial $r$ layers analogous to those chosen by the entire $m$ layers, or perhaps even superior? An analysis of the consistency or divergence of tokens selected by different layers, along with a comparison to tokens selected by SnapKV, would be highly appreciated.

---

> > > ### Author Response · Authors · 2024-11-25
> > > **Thank you and further reply to new concerns**
> > >
> > > We are glad that the reviewer likes our response! We sincerely thank you for your insightful feedback. We appreciate your time and we would like to answer your follow-up questions.
> > >
> > > ### Q1.1: GemFilter vs GemFilter-One-Run
> > >
> > > We would like to highlight that, both GemFilter and GemFilter-One-Run use exactly the same index set for the input, where the input set is generated from the layer 19 of Mistral Nemo. We update the revision Line 910 to highlight this.
> > >
> > > In detail,
> > > - In the first run, GemFilter will stop at layer 19 for long prompt computation and use the selected index set for the second run of short prompt computation.
> > > - GemFilter-One-Run will get the index set at layer 19, but not stop for its long prompt computation, i.e., it will finish all layer long prompt computation. Then, it uses the selected index set to evict the KV-Cache and starts iterative generation. There is no second run of short prompt computation.
> > >
> > > Thus, the only difference between them is that their RoPE embedding distance is different in the KV cache, i.e., GemFilter-One-Run may have a large RoPE distance between their tokens. We refer the reviewer to Line 908-917 in the second version revision for more details.
> > >
> > > ### Q1.2: An analysis of the consistency or divergence of tokens selected by different layers, along with a comparison to tokens selected by SnapKV, would be highly appreciated.
> > > Thank you for your brilliant suggestion! We have updated the corresponding experiment in Figure 11 and Line 964-1019 in the second version revision.
> > >
> > > In short, in Figure 11, we visualize the top-$k$ indices of each attention layer in GemFilter and SnapKV when using the Mistral Nemo 12B Instruct model and evaluating on Needle in a Haystack. Figure 11 shows that GemFilter is only focused on the needle information and recent information, while SnapKV focuses on a wide range of tokens, which may distract its attention. We can also see that GemFilter and SnapKV have very different selection mechanisms. We refer the reviewer to revision for more details.
> > >
> > > We hope our answer may address your new concerns.

---

> > > > ### Comment · Reviewer_bddG · 2024-11-26
> > > >
> > > > I appreciate the authors' efforts in addressing the concerns raised in the reviews, particularly the clarification about GemFilter-One-Run and the addition of Figure 11, which is indeed informative and helpful for understanding the methodology. One phenomenon that I notice in Figure 11 is the right window of $r$ is much sharper than expected (i.e., only 2 out of 40). This means that the selection of $r$ is more sensitive than it is claimed in Section 4.3, which claims performance remains robust in a much wider range. As pointed out by reviewer Np9u and rvyU, I also suspect that the selection of $r$ may be input-dependent. It is evident for the results in Table 2, which shows that no layer is consistently superior to others across all tasks.

---

> > > > > ### Author Response · Authors · 2024-11-27
> > > > > **Thank you and further reply to new concerns**
> > > > >
> > > > > We have plotted similar figures of index selection for the LLaMA 3.1 model and Phi 3.5 models in the third version of the revision, in Line 1049-1179.
> > > > >
> > > > > As we can see for the LLaMA 3.1 model and Phi 3.5, GemFilter has a wide range for layer index selection. However, we agree that GemFilter may be senestitity in the index selection on Mistral Nemo. All these results are also consistent with Figure 5, where Mistral Nemo has fewer 0 points than the other two models in Figure 5.
> > > > >
> > > > > On the other hand, in Needle in a haystack and Longbench, we always use layer 19 for Mistral Nemo and gain a good performance overall tasks. This filter layer is quite robust to all tasks in Table 1.
> > > > >
> > > > > We are also interested in the mechanism of the filter layer. We will continue to analyze and understand it in our follow-up works. Thank you again for your valuable suggestions!

---

> > > > > > ### Comment · Reviewer_bddG · 2024-11-30
> > > > > >
> > > > > > I would like to express my appreciation for the authors' efforts in addressing my concerns, which have significantly improved the clarity and addressed the concerns I raised in my previous review. I am satisfied with the authors' responses and the latest results. Thanks for the hard work and dedication to this research.

---

> > > > > > > ### Author Response · Authors · 2024-11-30
> > > > > > > **Thank you**
> > > > > > >
> > > > > > > We are glad that our response fully solves all your previous concerns. We sincerely thank the reviewer for constructive feedback and for helping us improve the quality of the draft. On the other hand, we hope to know any points we could do to improve the paper further and get a positive score from the reviewer. We appreciate your time and suggestions!

---

### Author Response · Authors · 2024-11-22
**Global Response**

We gratefully thank all reviewers for their valuable and constructive feedback.

We appreciate that all reviewers agree that our paper is interesting and focuses on a timely and critical problem. We are encouraged that reviewers bddG, rvyU, and Np9u recognize our method is simple yet effective and has promising results with good accuracy and memory/running time savings compared to state-of-the-art methods. We are glad that reviewers bddG and FAs6 think our draft is clear and easy to follow. Reviewers bddG and FAs6 also value the novelty of our methods, and reviewer FAs6 highlights that the flexible choice of filter layer of our method is quite useful in practice, avoiding extensive hyperparameter tuning.

We have updated a **revision** for our draft. We summarize all the major updates (in brown color) we made. All line numbers in the rebuttal correspond to the revised version.
- Line 157-160: We add a brief introduction of MInference [1], an important baseline.
- Line 174-179: We add a brief introduction of LLMLingua [2], an important baseline.
- Line 378-407 & 419-424 & 430-431: We update Table 1 with the above two baselines and add related discussion.
- Line 476-485 & 860-962: We add two more ablation studies.
- Line 121-122 & 489-490: We add the setting for the evaluation of running time and memory consumption.

Then, we will cover some questions that reviewers commonly ask.

### Missing baselines
Per the reviewers’ request, we add two more baselines, MInference [1] and LLMLingua [2], in Table 1 and the corresponding discussion in the revision Line 378-407 & 419-424 & 430-431. We can see that MInference has compatible performance with SnapKV, while it requires offline to determine the best attention pattern, which cannot save the prompt computation phase running time. We can see that although LLMLingua achieves a good comparison rate, the performance may not be satisfactory. Thus, GemFilter outperforms LLMLingua and has a performance that is compatible with MInference. Also, GemFilter is faster than MInference in the prompt computation phase.

We directly follow the official GitHub of MInference and LLMLingua for implementation. We only evaluate MInference on LLaMA 3.1 8B Instruct as it is not supported on the Mistral Nemo 12B Instruct and Phi 3.5 Mini 3.8B Instruct currently.

### Reference

[1] Jiang, H., Li, Y., Zhang, C., Wu, Q., Luo, X., Ahn, S., ... & Qiu, L. (2024). Minference 1.0: Accelerating pre-filling for long-context llms via dynamic sparse attention. NeurIPS’24.

[2] Jiang, H., Wu, Q., Lin, C. Y., Yang, Y., & Qiu, L. (2023). Llmlingua: Compressing prompts for accelerated inference of large language models. EMNLP’23.

---

### Author Response · Authors · 2024-11-25
**Further update in revision**

Per reviewer bddG and Np9u valuable suggestions, we update the second version revision with two new experiments. We summarize major updates (in blue color) we made.
- Line 910: We clarify that GemFilter-One-Run and GemFilter share exactly the same index set.
- Line 964-1019: We show the index selection difference between Gemfilter and SnapKV.
- Line 1020-1047: We report the performance of different methods on the LongBench QA task using LLaMA 3.1 8B Instruct and its official LLaMA Chat template.

We thank all reviewers for their constructive feedback and for helping us improve the draft.

---

### Author Response · Authors · 2024-11-27
**Third version revision update**

We made a third revision, where the update is in the color purple (we also keep our old updates). In Line 1049-1179, we show the index selection difference between Gemfilter and SnapKV on the LLaMA 3.1 model and Phi 3.5 models.

---

### Meta-Review · Area_Chair_LBgU · 2024-12-20

**Metareview:**

The paper proposes a novel approach, GemFilter, designed to accelerate inference and reduce memory consumption in large language models (LLMs) by utilizing early layers of the model to identify and compress relevant input tokens. This method aims to address the long-context bottleneck by filtering out unnecessary tokens early in the process, significantly decreasing the context size for subsequent layers. The authors present empirical results showing that GemFilter achieves a 2.4x speedup and a 30% reduction in GPU memory usage compared to state-of-the-art (SOTA) methods, such as standard attention and SnapKV/H2O. The method is claimed to be training-free, broadly applicable to various LLMs, and interpretable, with the ability to inspect the selected input tokens.

The strengths of the paper lie in its novel approach to tackling the long-context problem, a critical issue for LLMs. The proposed method demonstrates substantial empirical improvements, especially in terms of speed and memory efficiency, which are key for practical deployment. Additionally, the simplicity of GemFilter and its broad applicability across different LLM architectures make it a promising solution. The interpretability aspect is also noteworthy, as it allows users to inspect the token selection process, which is often opaque in large models. Overall, the paper is well-written and presents a clear and convincing argument for the effectiveness of the proposed method.

Despite these strengths, there are a number of unresolved issues. Reviewers raised concerns about the lack of comparison to other prompt compression methods, particularly LongLLMLingua, which would help place the contribution in a broader context. Furthermore, the authors did not provide an ablation study to isolate the contributions of token selection versus re-computation, leaving some ambiguity around the specific factors driving performance improvements. While the method shows promise, the lack of such detailed analysis limits the overall impact and understanding of the approach. Despite the authors' attempts to address the concerns raised by the reviewers in their rebuttal, the key issues remained unresolved.

Given the highly competitive nature of the ICLR 2025 submission process, the paper’s overall contribution does not stand out enough to justify acceptance. While the proposed method is interesting and offers practical benefits, the lack of sufficient comparison, theoretical insight, and detailed analysis makes it less competitive compared to other submissions. Consequently, the decision is to reject the paper.

**Additional Comments On Reviewer Discussion:**

During the rebuttal period, the reviewers raised several points, primarily focusing on the lack of comparison with prompt compression methods and the unclear attribution of performance improvements. One reviewer highlighted the need for a more detailed comparison with LongLLMLingua to better position GemFilter within existing optimization techniques. Another reviewer pointed out the absence of a thorough ablation study isolating the effects of token selection versus re-computation.

The authors addressed these concerns by providing additional comparisons, including a more in-depth discussion of LongLLMLingua. They also conducted an ablation study to clarify the contributions of token selection and re-computation. While the additional comparisons and the ablation study were helpful, they were not considered sufficient by the reviewers. The reviewers still felt the paper lacked the necessary depth and clarity, particularly in terms of theoretical insights and the broader applicability of the method. As a result, despite the authors’ efforts, the paper did not fully satisfy the reviewers' expectations, which contributed to the final decision to reject.

---

### Decision · Program_Chairs · 2025-01-22

Reject